# Diffusion Federated Dataset

**Seok-Ju Hahn**[1]
Argonne National Laboratory
hahns@anl.gov

**Junghye Lee**[2]
Seoul National University
junghye@snu.ac.kr

## Abstract

Diffusion models have demonstrated decent generation quality, yet their deployment in federated learning scenarios remains challenging. Due to data heterogeneity and a large number of parameters, conventional parameter averaging schemes often fail to achieve stable collaborative training of diffusion models. We reframe collaborative synthetic data generation as a cooperative sampling procedure from a mixture of decentralized distributions, each encoded by a pre-trained local diffusion model. This leverages the connection between diffusion and energy-based models, which readily supports compositional generation thereof. Consequently, we can directly obtain refined synthetic dataset, optionally with differential privacy guarantee, even without exchanging diffusion model parameters. Our framework reduces communication overhead while maintaining the generation quality, realized through an unadjusted Langevin algorithm with a convergence guarantee.

## 1 Introduction

Federated learning (FL [1]) enables clients (i.e., data owners) to collaboratively train a statistical model by exchanging locally updated parameters with a central server over iterative communication rounds, thereby preserving data privacy. While this *model-centric FL* paradigm is well-established, sharing public data can substantially enhance FL performance by mitigating statistical heterogeneity arising from non-independent and identically distributed (non-IID) local data distributions [1–4]. For instance, public or synthetic datasets can homogenize disparate local distributions and serve as direct signals for server-side pretraining. This facilitates client-side transfer learning or data augmentation, both improve the overall utility of FL.

While these *data-centric FL scheme* offers clear advantages over purely model-centric approaches, there remain challenges. First, curating public datasets is often infeasible in a real-world FL system. Although generation of synthetic data via the collaborative training of generative models is a viable alternative, it is challenging in FL settings. For example, generative adversarial networks (GANs [5]) suffer from training instabilities and suboptimal sample quality, which are exacerbated in FL by statistical heterogeneity [6–8]. Even advanced diffusion models [9–12] incur substantial computational and communication overheads due to their large parameter sizes and fine-grained optimization requirements. Thus, effective synthetic data generation methods for FL constitute a critical yet underexplored research area. Specifically in cross-silo FL settings, clients often have a limited number of samples (e.g., hospitals or enterprises with small datasets). In this sample-limited condition, generating synthetic data becomes critical as it can complement scarce and disparate local dataset with high-fidelity synthetic samples. Thereby, this directly addresses the statistical heterogeneity problem in federated environments.

In this work, we redefine federated synthetic data generation as a *collaborative sampling process* from a mixture of heterogeneous and inaccessible local distributions. By modeling each local distribution with a client-side diffusion model, we enable efficient compositional sampling from them, leveraging

---

Code is available at: https://github.com/vaseline555/DfD

39th Conference on Neural Information Processing Systems (NeurIPS 2025).

energy-based interpretations of diffusion models [13] and the mixture-of-experts paradigm [14]. The sampling is embarrassingly simple, through the unadjusted Langevin algorithm (ULA [15, 16]). Building on these, we introduce DfD (Diffusion-federated Dataset), a cooperative inference framework that generates synthetic data through *sampling directly from a mixture distribution*, eschewing traditional model averaging. DfD advances federated synthetic data generation as follows:

- We propose a novel view on federated synthetic data generation as cooperative sampling from individually trained diffusion models, without necessitating the exchange of model parameters.

- Through energy-based parameterization and compatibility of ULA with the diffusion reverse process, we refine the connections between diffusion models and energy-based models (EBMs [17]). We also derive the optimal step size and non-asymptotic distributional convergence for DfD.

- We empirically validate fidelity and utility of synthetic dataset from DfD under non-IID conditions, optionally with formal privacy guarantees, addressing key needs in cross-silo FL scenarios.

## 2 Related Works

**Synthetic Data in FL.** FL often struggles with slow convergence when the client's local private data sets differ significantly, a common challenge known as statistical heterogeneity or the non-IID problem [1, 4]. This issue is critical in that the central server cannot directly access or adjust these heterogeneous local datasets to align their disparate optimization trajectories. Most prior work has addressed this through *model-centric* approaches, such as local update regularization [18–21], modified central aggregation schemes [22–27], or personalization [28–30].

While effective, a complementary *data-centric* perspective still remains underexplored. These include sharing additional server-side public data [2, 31–33], using indiscernible auxiliary representations [34–40], or leveraging a generative model to obtain plausible synthetic data [41–50]. These provide clients with a proxy for global distribution, which directly mitigates the non-IID problem and improves convergence [51]. Notably, as studied in [2], sharing only a small portion of public data can significantly boost FL performance, though acquiring such data is nontrivial in practice.

Hence, synthetic data is widely used with generative models in e.g., healthcare [52–57]. However, current synthetic data generation methods in FL, including real-world applications, mostly resort to GANs [58] (optionally with privacy guarantee [59–62]), which suffer from subpar generation quality and optimization instability due to their adversarial training scheme (e.g., mode collapse [6–8]).

**Diffusion Models in FL.** Diffusion models [11, 12], such as Denoising Diffusion Probabilistic Models (DDPMs [9]), have offered a superior generation quality training stability, compared to other generative models, e.g., GANs. [10]. Although promising, their adoption in FL is challenging and sometimes even prohibitive due to high computational costs and large model sizes. Thus, current methods suffer from significant communication overhead [63], poor scalability to high-resolution data [64], and even require retraining of local models [65] or data sharing [66] due to non-IID problem. In addition, the inherent loss design of diffusion models, which depend on multiple time-steps, also requires frequent parameter exchanges during training, making them difficult to adopt in FL [66, 67].

Our framework detours by directly generating samples from an inaccessible mixture of heterogeneous local distributions, encoded by locally-trained diffusion models. This is rooted in exploiting the connection of diffusion models to energy-based models (EBMs [17]), which estimate unnormalized probability densities through their gradients with respect to inputs (i.e., *scores* [68]). This intriguing connection enables easy compositional sampling, which can be viewed as sampling from a mixture-of-experts [14], even without accessing model parameters. As a result, DfD offers an efficient and scalable solution for adopting diffusion models in federated synthetic data generation.

## 3 Preliminaries

### 3.1 Diffusion Models

Diffusion models aim to encode data distribution $p_{\text{data}}(\boldsymbol{x})$ by learning transition from noise-perturbed data $\{\boldsymbol{x}_t\}_{t=1}^T$, where $\boldsymbol{x}_T \sim \mathcal{N}(\boldsymbol{0}_d, \mathbf{I}_d)$, into its clean original counterpart, $\boldsymbol{x}_0 \sim p_{\text{data}}(\boldsymbol{x}) \equiv q(\boldsymbol{x}_0)$ through paired forward and backward processes. Specifically, Gaussian diffusion defines a Markov

chain joint distribution $q(\boldsymbol{x}_0, ..., \boldsymbol{x}_T) = q(\boldsymbol{x}_0) \prod_{t=1}^{T} q(\boldsymbol{x}_t | \boldsymbol{x}_{t-1})$, where the *forward process* is defined by incrementally adding Gaussian noise over $t \in [T]$ as $q(\boldsymbol{x}_t | \boldsymbol{x}_{t-1}) = \mathcal{N}(x_t; \sqrt{\alpha_t} \boldsymbol{x}_{t-1}, \beta_t \mathbf{I}_d)$. Note that $d$ is the data dimension, $0 < \beta_t \leq 1$ and $\alpha_t = 1 - \beta_t$ are noise constants. The *reverse process*, typically parameterized by a deep network with $\boldsymbol{\theta}$, approximates $p_{\boldsymbol{\theta}}(\boldsymbol{x}_{t-1} | \boldsymbol{x}_t)$, in order to progressively denoise from the Gaussian noise $\boldsymbol{x}_T$ into the original data $\boldsymbol{x}_0$. With sufficiently small $\beta_t$, each transition of reverse process approximately follows Gaussian [11]. This allows:

$$p_{\boldsymbol{\theta}}(\boldsymbol{x}_{t-1} | \boldsymbol{x}_t) = \mathcal{N}\Big(\boldsymbol{x}_{t-1}; \underbrace{\frac{1}{\sqrt{\alpha_t}}\Big(\boldsymbol{x}_t - \frac{\beta_t}{\sqrt{1 - \bar{\alpha}_t}}\boldsymbol{\epsilon}_{\boldsymbol{\theta}}(\boldsymbol{x}_t, t)\Big)}_{=:\boldsymbol{\mu}_{\boldsymbol{\theta}}(\boldsymbol{x}_t, t)}, \tilde{\beta}_t \mathbf{I}_d\Big), \tag{1}$$

where $\bar{\alpha}_t, \tilde{\beta}_t$ are some transformations of $\beta_t, \forall t \in [T]$, following the configurations of [9] (see also Appendix C.1).

Eventually, the parameterized deep network needs to predict $\boldsymbol{\epsilon}_{\boldsymbol{\theta}}(\boldsymbol{x}_t, t)$ as a mapping $\boldsymbol{\epsilon}_{\boldsymbol{\theta}} : \mathbb{R}^d \times [T] \to \mathbb{R}^d$. Note that diffusion models ensure the analytic conversion from the original to the perturbed data at any timestep $t \in [T]$ [9]:

$$\boldsymbol{x}_t = \sqrt{\bar{\alpha}_t}\boldsymbol{x}_0 + \sqrt{1 - \bar{\alpha}_t}\boldsymbol{\epsilon}, \quad \boldsymbol{\epsilon} \sim \mathcal{N}(\boldsymbol{0}_d, \mathbf{I}_d). \tag{2}$$

Using this property, we can optimize with composite loss [9] as $\mathcal{L}(\boldsymbol{\theta}) = \sum_{t=1}^{T} \mathcal{L}(\boldsymbol{\theta}, t)$, where

$$\mathcal{L}(\boldsymbol{\theta}, t) = \mathbb{E}_{\boldsymbol{x}_0 \sim p_{\text{data}}(\boldsymbol{x}), \boldsymbol{\epsilon} \sim \mathcal{N}(\boldsymbol{0}_d, \mathbf{I}_d)} \left[ \|\boldsymbol{\epsilon} - \boldsymbol{\epsilon}_{\boldsymbol{\theta}}(\boldsymbol{x}_t, t)\|_2^2 \right]. \tag{3}$$

By minimizing this objective, diffusion models are capable of generating high-quality samples by constructing $\boldsymbol{\mu}_{\boldsymbol{\theta}}(\boldsymbol{x}_t, t)$ from their prediction $\boldsymbol{\epsilon}_{\boldsymbol{\theta}}(\boldsymbol{x}_t, t)$ and progressively denoising from $\boldsymbol{x}_T \sim \mathcal{N}(\boldsymbol{0}_d, \mathbf{I}_d)$ to $\boldsymbol{x}_0$ over $t = T - 1, ..., 1$, using Eq. (1). We refer to Appendix A for detailed derivations.

### 3.2 Energy-based Interpretation of Diffusion Models

Diffusion models have an intriguing connection with EBMs [12, 13]. EBMs [17] define an unnormalized probability density as:

$$p_{\boldsymbol{\theta}}(\boldsymbol{x}) = \frac{\exp\left(-\lambda f_{\boldsymbol{\theta}}(\boldsymbol{x})\right)}{Z_{\boldsymbol{\theta}}}, \tag{4}$$

where EBMs forgo modeling of the normalizing constant $Z_{\boldsymbol{\theta}} = \int_{\boldsymbol{x} \in \mathcal{X}} \exp\left(-\lambda f_{\boldsymbol{\theta}}(\boldsymbol{x})\right) \mathrm{d}\boldsymbol{x}$. We define $f_{\boldsymbol{\theta}} : \mathbb{R}^d \to \mathbb{R}$ as an *energy function* with parameter $\boldsymbol{\theta} \in \mathbb{R}^p$, scale factor $\lambda \in \mathbb{R}^+$ and $\nabla_{\boldsymbol{x}} \log p_{\boldsymbol{\theta}}(\boldsymbol{x}) = -\lambda \nabla_{\boldsymbol{x}} f_{\boldsymbol{\theta}}(\boldsymbol{x})$ as a *score*. Note that we have $d \ll p$ if we choose deep networks, which are typically overparameterized.

The abstention of modeling normalizing constant prevents exact likelihood computation. To address the issues that arise from this design, *denoising score matching* [69] has been proposed to minimize the Fisher divergence between the model's score and that of a noise-perturbed data distribution, i.e., $q(\boldsymbol{x}_\sigma) = \int_{\boldsymbol{x} \in \mathcal{X}} q_\sigma(\boldsymbol{x}_\sigma | \boldsymbol{x}) p_{\text{data}}(\boldsymbol{x}) \mathrm{d}\boldsymbol{x}$. Note here that $\sigma$ is a noise variance and the perturbation is given as $\boldsymbol{x}_\sigma = \boldsymbol{x} + \sigma\boldsymbol{\epsilon}, \boldsymbol{\epsilon} \sim \mathcal{N}(\boldsymbol{0}_d, \mathbf{I}_d)$. Building on these, the denoising score matching objective is:

$$\mathcal{L}(\boldsymbol{\theta}, \sigma) = \mathbb{E}_{\boldsymbol{x}_\sigma \sim q(\boldsymbol{x}_\sigma | \boldsymbol{x}), \boldsymbol{x} \sim p_{\text{data}}(\boldsymbol{x})} \left[ \|\nabla_{\boldsymbol{x}_\sigma} \log q(\boldsymbol{x}_\sigma | \boldsymbol{x}) - \nabla_{\boldsymbol{x}_\sigma} \log p_{\boldsymbol{\theta}}(\boldsymbol{x}_\sigma)\|^2 \right]. \tag{5}$$

This is equivalent (up to a constant) to:

$$\sigma^2 \mathcal{L}(\boldsymbol{\theta}, \sigma) = \mathbb{E}_{\boldsymbol{x}_\sigma \sim q(\boldsymbol{x}_\sigma | \boldsymbol{x}), \boldsymbol{\epsilon} \sim \mathcal{N}(\boldsymbol{0}_d, \mathbf{I}_d)} \left[ \|\boldsymbol{\epsilon} - \sigma\lambda \nabla_{\boldsymbol{x}_\sigma} f_{\boldsymbol{\theta}}(\boldsymbol{x}_\sigma)\|^2 \right]. \tag{6}$$

Interestingly, the objective of diffusion models in Eq. (3) aligns with the scaled objective above [13], with following connection (along with replacing $\sigma$ into $\sigma_t$):

$$\nabla_{\boldsymbol{x}_{\sigma_t}} \log p_{\boldsymbol{\theta}}(\boldsymbol{x}_{\sigma_t}) = -\lambda \nabla_{\boldsymbol{x}_{\sigma_t}} f_{\boldsymbol{\theta}}(\boldsymbol{x}_{\sigma_t}) \equiv -\frac{\boldsymbol{\epsilon}_{\boldsymbol{\theta}}(\boldsymbol{x}_t, t)}{\sigma_t}. \tag{7}$$

Note that this connection to EBMs can be further concretized for diffusion models by a specific choice of the *energy-based parameterization* introduced in following Section 3.4. It should also be noted that this explicit connection allows using a sampler for diffusion models, e.g., ULA. We defer all detailed derivations in this section to Appendix B.

## 3.3 Federated Synthetic Data Generation by Sampling from a Mixture Distributions

The ULA follows a discretized Langevin diffusion process [15] and enables sampling from a target distribution $p(\boldsymbol{x})$ with its score $\nabla_{\boldsymbol{x}} \log p(\boldsymbol{x})$, by iteratively updating from $\boldsymbol{x}_T \sim \mathcal{N}(\mathbf{0}_d, \mathbf{I}_d)$ using:

$$\boldsymbol{x}_{t-1} = \boldsymbol{x}_t + \eta_t \nabla_{\boldsymbol{x}_t} \log p(\boldsymbol{x}_t) + \sqrt{2\eta_t}\boldsymbol{z}_t, \quad \boldsymbol{z}_t \sim \mathcal{N}(\mathbf{0}_d, \mathbf{I}_d), \tag{8}$$

where $\eta_t \geq 0$ is a step size, and it ensures $\boldsymbol{x}_0 \sim p(\boldsymbol{x})$ [70]. We denote the notation of decreasing timesteps as $t = T, ..., 1$ for the compatibility with the diffusion reverse process.

In FL setup, we have $K$ clients each having private dataset $\mathcal{D}_i$. Then, a target distribution is naturally defined as a mixture of local distributions: $p^\star(\boldsymbol{x}) = \sum_{i=1}^{K} w_i p_i(\boldsymbol{x})$, where $p_i(\boldsymbol{x})$ represents unknown local distribution of $\mathcal{D}_i$ from $i$-th client and $w_i \geq 0$ is a mixing coefficient satisfying $\sum_{i=1}^{K} w_i = 1$ (e.g., $w_i = 1/K$ if uniform weighting). To generate samples from the mixture of local distributions, what we need to estimate the *global score* $\nabla_{\boldsymbol{x}} \log p^\star(\boldsymbol{x})$ defined as follows:

$$\nabla_{\boldsymbol{x}} \log p^\star(\boldsymbol{x}) = \sum_{i=1}^{K} \tilde{w}_i \nabla_{\boldsymbol{x}} \log p_{\boldsymbol{\theta}_i}(\boldsymbol{x}), \quad \tilde{w}_i = \frac{w_i \exp(-\lambda f_{\boldsymbol{\theta}_i}(\boldsymbol{x}))}{\sum_{j=1}^{K} w_j \exp(-\lambda f_{\boldsymbol{\theta}_j}(\boldsymbol{x}))}, \tag{9}$$

where $\tilde{w}_i$ is derived from $p_{\boldsymbol{\theta}_i}(\boldsymbol{x}) \propto \exp(-\lambda f_{\boldsymbol{\theta}_i}(\boldsymbol{x}))$ due to Eq. (4). Note that it directly supports embarrassingly parallel computation across clients, aligning well with FL settings. In detail, the estimation of the global score $\nabla_{\boldsymbol{x}} \log p^\star(\boldsymbol{x})$ is available as long as we have both i) *local scores* $\nabla_{\boldsymbol{x}} \log p_{\boldsymbol{\theta}_i}(\boldsymbol{x})$ and ii) *energies* (unnormalized density values) $\exp(-\lambda f_{\boldsymbol{\theta}_i}(\boldsymbol{x}))$ of each client.

However, diffusion models do not explicitly provide $f_{\boldsymbol{\theta}_i}(\boldsymbol{x})$ in its inherent design. This can be easily addressed using energy-based parameterization described in the following section.

## 3.4 Energy-based Parameterization of Diffusion Models

To implement ULA to directly sample from a mixture of local distributions, we should estimate the energies $p_{\boldsymbol{\theta}_i}(\boldsymbol{x}) \propto \exp(-\lambda f_{\boldsymbol{\theta}_i}(\boldsymbol{x}))$ for $\tilde{w}_i$ in Eq. (9). Since diffusion models lack explicit density function, prior arts proposed to approximate them by defining $f_{\boldsymbol{\theta}}(\boldsymbol{x})$ using an energy-based $\ell_2$ parameterization trick [13, 71]. (We refer to Section D of [13] for details on other tricks)

**Definition 3.1** (energy-based $\ell_2$ parameterization [13]). The energy function of a diffusion model is approximated as $f_{\boldsymbol{\theta}}(\boldsymbol{x}_t, t) = \frac{1}{2}\|\boldsymbol{\epsilon}_{\boldsymbol{\theta}}(\boldsymbol{x}_t, t)\|_2^2$, where $\boldsymbol{\epsilon}_{\boldsymbol{\theta}}(\boldsymbol{x}_t, t)$ is a prediction of a diffusion model.

Having this energy function, we can now define scores of diffusion models and obtain the global score in Eq. (9), accordingly. Unfortunately, this parameterization requires a modification in training of diffusion models, and this often yields subpar generation quality [13, 71].

# 4 Proposed Method

## 4.1 Refined Energy-based Parameterization

To detour the modification in training, we start from the notion of *well-trained diffusion models*.

**Definition 4.1** (Well-trained diffusion model). A diffusion model is well-trained if, through minimization of the objective in Eq. (3), its noise prediction satisfies $\boldsymbol{\epsilon}_{\boldsymbol{\theta}}(\boldsymbol{x}_t, t) \approx \boldsymbol{\epsilon} \sim \mathcal{N}(\mathbf{0}_d, \mathbf{I}_d)$.

**Remark 4.2.** Note that this captures the empirical observation that sufficiently trained diffusion models accurately predict added noise. In addition, thanks to $\boldsymbol{\epsilon} = (\boldsymbol{x}_t - \sqrt{\bar{\alpha}_t}\boldsymbol{x}_0)/\sqrt{1 - \bar{\alpha}_t}$ from Eq. (2), a well-trained diffusion model readily satisfies that $\nabla_{\boldsymbol{x}_t}\boldsymbol{\epsilon}_{\boldsymbol{\theta}}(\boldsymbol{x}_t, t) \approx \nabla_{\boldsymbol{x}_t}\boldsymbol{\epsilon} = \frac{1}{\sqrt{1-\bar{\alpha}_t}}\mathbf{I}_d$.

With these, we can now approximate the score of well-trained diffusion models as follows:

$$\begin{aligned} \nabla_{\boldsymbol{x}_t} \log p_{\boldsymbol{\theta}}(\boldsymbol{x}_t, t) &= -\lambda \nabla_{\boldsymbol{x}_t} f_{\boldsymbol{\theta}}(\boldsymbol{x}_t, t) \\ &= -\lambda \boldsymbol{\epsilon}_{\boldsymbol{\theta}}(\boldsymbol{x}_t, t)^\mathsf{T} \nabla_{\boldsymbol{x}_t} \boldsymbol{\epsilon}_{\boldsymbol{\theta}}(\boldsymbol{x}_t, t) \\ &\approx -\frac{\lambda}{\sqrt{1 - \bar{\alpha}_t}} \boldsymbol{\epsilon}_{\boldsymbol{\theta}}(\boldsymbol{x}_t, t), \end{aligned} \tag{10}$$

where the first equality is a direct result from Eq. (4), the second equality is due to Definition 3.1, and the last approximation is from Definition 4.1. Notably, this matches Eq. (7) if $\sigma_t = \sqrt{1 - \bar{\alpha}_t}/\lambda$. To

summarize, the *refined* energy-based $\ell_2$ reparameterization provides an unnormalized density and a score of well-trained diffusion models as:

$$p_{\boldsymbol{\theta}}(\boldsymbol{x}_t, t) \propto \exp\left(-\frac{\lambda}{2}\|\boldsymbol{\epsilon}_{\boldsymbol{\theta}}(\boldsymbol{x}_t, t)\|_2^2\right), \quad \nabla_{\boldsymbol{x}_t} \log p_{\boldsymbol{\theta}}(\boldsymbol{x}_t, t) = -\frac{\lambda}{\sqrt{1 - \bar{\alpha}_t}} \boldsymbol{\epsilon}_{\boldsymbol{\theta}}(\boldsymbol{x}_t, t), \quad (11)$$

for all timesteps $t \in [T]$. With these, *no modification to the training* of diffusion models is required.

## 4.2 `DfD`: Cooperative Diffusion Models Inference Framework for Synthetic Dataset

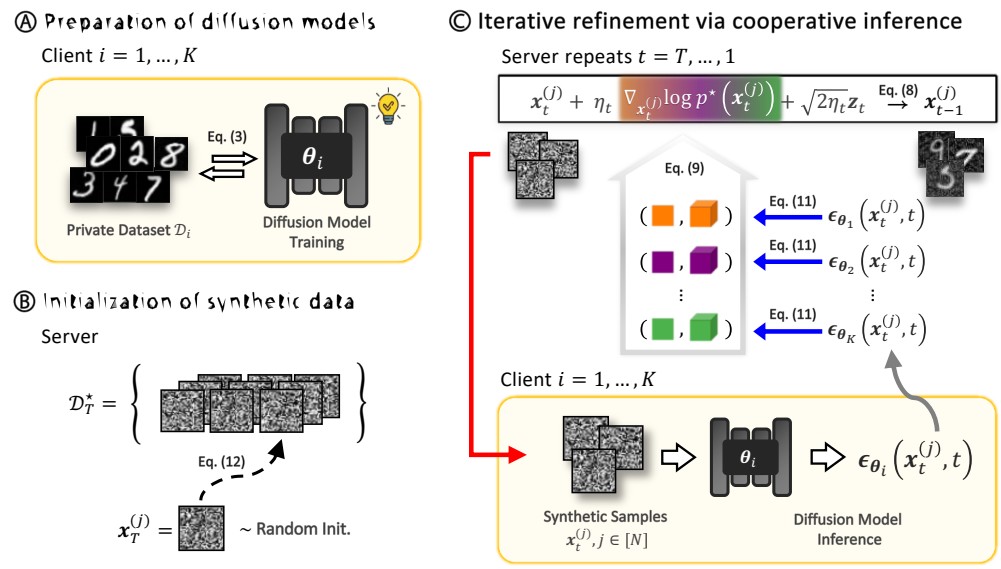

Figure 1: **Overview of `DfD`.** Ⓐ Clients independently train diffusion models to be well-trained with Eq. (3). Ⓑ The server randomly initializes synthetic dataset per Eq. (12). Ⓒ The server requests (➡) inference on synthetic dataset to all clients, receives (➡) predictions $\boldsymbol{\epsilon}_{\boldsymbol{\theta}_i}(\boldsymbol{x}_t^{(j)}, t), \forall i \in [K], j \in [N]$, transforms (➡) into energies (🟧, 🟪, 🟩) and scores (🟠, ⬤, 🟢) using Eq. (11), composes into global scores using Eq. (9), and refines synthetic dataset using ULA in Eq. (8) over $T$ steps.

Our proposed framework, `DfD`, generates synthetic data samples directly from a mixture of local distributions encoded by local diffusion models, independently trained on private and non-IID client datasets. An overview of the framework is provided in Figure 1 and the overall procedure of `DfD` (for the case of unconditional generation) is described in Algorithm 1.

A key innovation of `DfD` is its ability to leverage locally trained diffusion models, avoiding repetitive local updates along with the exchange of model parameters. This is achieved by exchanging the predictions of well-trained diffusion models instead, and the models are prepared by each client before cooperative inference begins. These predictions are iteratively collected and transformed into energies and scores at the server, to construct a global score in Eq. (9).

Ⓐ **Preparation of diffusion models.** Each client $i$ trains its own diffusion model on its private dataset $\mathcal{D}_i$ by minimizing Eq. (3), to obtain a *well-trained* model as in Definition 4.1. The model can be unconditional, predicting $\boldsymbol{\epsilon}_{\boldsymbol{\theta}_i}(\boldsymbol{x}_t, t)$, or conditional on label $\boldsymbol{y}$ (e.g., attributes or classes), predicting $\boldsymbol{\epsilon}_{\boldsymbol{\theta}_i}(\boldsymbol{x}_t, \boldsymbol{y}, t)$. Note that the dimension of predictions is equal to that of inputs, which is significantly smaller than the model parameter size. In addition, local pre-training can occur asynchronously, and clients may optionally apply differential privacy (DP) mechanisms [72].

---

**Algorithm 1** `DfD`: Cooperative Diffusion Models Inference Framework for Synthetic Dataset

---
1: **Require:** number of clients $K$, synthetic dataset size $N$, communication rounds $T$
2: **Procedure:**
3:     **All clients** $i \in [K]$ prepare a well-trained diffusion model $\boldsymbol{\theta}_i \in \mathbb{R}^p$ using $\mathcal{D}_i$ with Eq. (3).
4:     Server initializes $N$ samples in Eq. (12) to have $\mathcal{D}_T^\star$.
5:     **for** $t = T, ..., 1$ **the server**
6:         Requests inference to all clients in parallel on $\mathcal{D}_t^\star$.
7:         Receives predictions $\{\boldsymbol{\epsilon}_{\boldsymbol{\theta}_i}(\boldsymbol{x}_t^{(j)}, t) \in \mathbb{R}^d \mid i \in [K], j \in [N]\}$.
8:         Transforms predictions into energies and scores with Eq. (11).
9:         Computes global scores for all samples with Eq. (9).
10:       Updates synthetic dataset into $\mathcal{D}_{t-1}^\star$ using ULA in Eq. (8).
11:     **end for**
12: **Return:** $\mathcal{D}_0^\star$

---

Ⓑ **Initialization of synthetic data.** The central server randomly initializes $N$ synthetic data samples $\mathcal{D}_T^\star = \{\boldsymbol{x}_T^{(j)}\}_{j=1}^N$ (or $\mathcal{D}_T^\star = \{(\boldsymbol{x}_T^{(j)}, \boldsymbol{y}^{(j)})\}_{j=1}^N$) as:

$$\boldsymbol{x}_T^{(j)} \sim \mathcal{N}(\mathbf{0}_d, \mathbf{I}_d), \qquad \text{(if conditional)} \quad \boldsymbol{y}^{(j)} \sim \text{Categorical}\left(C^{-1}\mathbf{1}_C\right), \tag{12}$$

where $C$ is the number of conditions (e.g., classes, attributes) encoded by labels. The synthetic dataset size $N$ is determined based on communication constraints, where $N$ can be set much smaller than the required parameter size of diffusion models, e.g., $N \ll \max_i \dim(\boldsymbol{\theta}_i)$.

Ⓒ **Iterative refinement via cooperative inference.** For each communication round $t = T, ..., 1$, the central server sends the current synthetic dataset to all clients and requests predictions from their diffusion models. With these predictions, the server computes energies and scores of each client using Eq. (11). The server then constructs global scores using Eq. (9) and refines the server-side synthetic dataset using ULA, as in Eq. (8). Note that it can be extended to the conditional case by simply incorporating $\boldsymbol{y}^{(j)}$ in this step. At the end, the server obtains a refined synthetic dataset, $\mathcal{D}_0^\star$.

### 4.3 Theoretical Analysis

The ULA is the main workhorse of `DfD` as it relies on energy-based parameterization to sample from a mixture of local distributions using global scores in Eq. (9). Hence, we must carefully select the step size, denoted by $\eta_t$, to ensure that the `DfD` correctly settles at the target mixture distribution. We theoretically derive the step size guidance in two steps: ⓐ verification of the compatibility of ULA with diffusion reverse process, and ⓑ analysis of non-asymptotic convergence behavior of ULA to the target distribution in KL divergence [70]. We defer all proofs in Appendix C.

ⓐ **Compatibility of ULA with diffusion reverse process.** `DfD` resort to diffusion models as main components. Thus, we begin with the successful diffusion reverse process and transplant its key success factor into the ULA to ensure compatibility. Interestingly, we find that *non-expansiveness* w.r.t. $\ell_2$-norm is inherently encoded in the diffusion reverse process, and perceive it as a key factor.

**Lemma 4.3** (Non-expansiveness of diffusion reverse process)**.** *The diffusion reverse process in Eq.* (1) *preserves the squared $\ell_2$-norm of resulting iterates to be non-expansive, i.e.,* $\mathbb{E}[\|\boldsymbol{x}_{t-1}\|_2^2] \leq \mathbb{E}[\|\boldsymbol{x}_t\|_2^2]$.

Next, we proved that this property can be similarly induced for ULA under following conditions. This gives an explicit guidance for the choice of scale factor $\lambda$ in Eq. (4), which is used for the construction of energies and scores in Eq. (9).

**Lemma 4.4** (Non-expansiveness condition of ULA)**.** *ULA satisfies the non-expansiveness w.r.t. squared $\ell_2$-norm as* $\mathbb{E}[\|\boldsymbol{x}_{t-1}\|_2^2] \leq \mathbb{E}[\|\boldsymbol{x}_t\|_2^2]$, *for well-trained diffusion models with energy-based $\ell_2$ parameterization, if and only if $\eta_t \in [0, \frac{1}{2}]$ and $\lambda = 2$.*

ⓑ **Non-asymptotic convergence of ULA.** Though previous work heuristically adopted the naive resemblance of ULA with the diffusion reverse process to set the step size (i.e., simply setting $\eta_t = \beta_t$ while ignoring the scaling factor $\frac{1}{\sqrt{\alpha_t}}$) [71], this approach has no theoretical justification. Thus, we theoretically derive a ULA step size and the convergence guarantee toward a target mixture distribution under KL divergence, with acceptable assumptions provided in Appendix C.

**Theorem 4.5** (Convergence guarantee of `DfD`). *Let $\tilde{p}_t$ be the evolving distribution of $\boldsymbol{x}_t \in \mathbb{R}^d$ from ULA and $p_t$ be the mixture of distributions encoded by diffusion models. For $\delta \geq \frac{18d\varsigma\eta_T(1-\bar{\alpha}_{T-t})}{\upsilon}$ and $\rho \in (0, \sqrt{3}/6)$, the iterates $\boldsymbol{x}_{T-t} \sim \tilde{p}^{\star}_{T-t}$ guarantee $\mathrm{D}_{\mathrm{KL}}(\tilde{p}_{T-t} \parallel p_{T-t}) < \delta$ after $t \geq \frac{1-\bar{\alpha}_{T-t}}{\upsilon} \log\left(\frac{2\,\mathrm{D}_{\mathrm{KL}}(\tilde{p}_T \parallel p_T)}{\delta}\right)$ steps with a step size $\eta_{T-t} \leq \min\left\{\frac{\upsilon\delta}{18d\varsigma(1-\bar{\alpha}_{T-t})}, \frac{\rho(1-\bar{\alpha}_{T-t})^p}{2^p}\right\}$.*

Note that this ensures `DfD` can sample from a mixture of inaccessible and heterogeneous distributions in a finite number of steps, *without access* to the local dataset $\mathcal{D}_i$ and the local model parameters $\boldsymbol{\theta}_i$.

## 4.4 Privacy Guarantee

**Definition 4.6** (($\epsilon, \delta$)-DP [72]). A mechanism $\mathcal{M}$ satisfies ($\epsilon, \delta$)-DP if, for any two neighboring datasets $\mathcal{D}$ and $\mathcal{D}'$ differing in one record, and for any output set $S$, $\Pr[\mathcal{M}(\mathcal{D}) \in S] \leq e^{\epsilon} \Pr[\mathcal{M}(\mathcal{D}') \in S] + \delta$, where $\epsilon > 0$ is the privacy budget and $\delta \geq 0$ is the failure probability.

The communicated signals in `DfD` are client predictions $\boldsymbol{\epsilon}_{\boldsymbol{\theta}_i}(\boldsymbol{x}^{(j)}_t, t)$ from a diffusion model trained on a private dataset $\mathcal{D}_i$. In FL, we typically use DP mechanism to protect sensitive information. Intriguingly, `DfD` can inherit DP guarantee as long as each client $i$ already trained its own diffusion model $\boldsymbol{\theta}_i$ to achieve ($\epsilon_i, \delta_i$)-DP, e.g., using DP-SGD [73].

**Theorem 4.7** (DP guarantee of `DfD`). *Assume all client datasets $\mathcal{D}_i$ are disjoint. If each client $i \in [K]$ trains a diffusion model $\boldsymbol{\theta}_i$ for ($\epsilon_i, \delta_i$)-DP given $\epsilon_i > 0$ and $\delta_i \geq 0$, the synthetic dataset $\mathcal{D}^{\star}_0$ generated by `DfD` compositely satisfies $(\max_i \epsilon_i, \max_i \delta_i)$-DP.*

*Proof.* As the server processes differentially private local predictions, the post-processing property of DP [74] also ensures that subsequent steps (i.e., global score computation, ULA updates) to preserve DP. The parallel composition theorem [75] provides a composite DP guarantee across clients with disjoint datasets with each other, in terms of the maximum privacy budget and failure probability. $\square$

# 5 Experimental Results

## 5.1 Setup

**Datasets.** We use three benchmark datasets: MNIST [76], CIFAR-10 [77], and CelebA [78], after resizing all inputs to have spatial dimension of $32\times32$. As each dataset has separate train & test folds, we use the train fold to split into client datasets, and set the test fold aside for server-side evaluation. We distribute the train fold of each dataset into $K = 10$ clients with three different non-IID conditions: i) Dirichlet distribution-based non-IID [79] for MNIST, ii) power-law distribution-based non-IID [21] for CIFAR-10, and iii) pathological non-IID [1] for CelebA.

To further simulate a convincing scenario in which a synthetic dataset should be procured (i.e., *data-limited settings*), we randomly sample local dataset to have a size of 300 on average, following the sample size configurations of the curated benchmark for the cross-silo FL setting [80].

**Baselines.** We compare with FL methods for generative models: `FedGAN` [42], `FedDiffuse` [63] and `PRISM` [67]. All clients are taking 10K steps in total for $T = 1000$ rounds: $E = 10$ local updates for all comparison methods, and $E = 10 \times 1,000 = 10,000$ local updates for `DfD` as it requires no update during communication rounds. The mini-batch size is set to $B = 32$, and the learning rates are tuned for all methods, and set to $c(1 - \bar{\alpha}_t)^p$ for $c > 0, p \geq 1$ for `DfD`.

**Evaluation Metrics.** We evaluate both fidelity and utility of the generated synthetic dataset. To evaluate the fidelity of synthetic data, we use the widely-used metrics for generative modeling: Fréchet Inception Distance (FID [81]), Precision & Recall (P&R [82]), and Density & Coverage (D&C [83]). To evaluate utility, we use an accuracy evaluated from a classifier trained at the central server using class-labeled synthetic dataset. We defer the specific experimental setup to Appendix D.

Table 1: Results on synthetic dataset quality.

| | | FID ↓ | P ↑ | R ↑ | D ↑ | C ↑ |
|---|---|---|---|---|---|---|
| MNIST | FedGAN [42] | **34.8486** | 0.4189 | 0.1240 | 0.1144 | 0.1378 |
| | FedDiffuse [63] | 49.5704 | 0.1842 | **0.7610** | 0.1145 | 0.3428 |
| | PRISM [67] | 36.7945 | 0.4223 | 0.1386 | 0.1639 | 0.1481 |
| | DfD | 37.7354 | **0.6224** | 0.3437 | **0.1816** | **0.3937** |
| CIFAR-10 | FedGAN [42] | 145.5668 | **0.6866** | 0.0221 | **0.4800** | 0.1221 |
| | FedDiffuse [63] | 78.3845 | 0.4142 | 0.2119 | 0.3731 | 0.2958 |
| | PRISM [67] | 330.8488 | 0.0875 | 0.0077 | 0.0334 | 0.0368 |
| | DfD | **59.9761** | 0.5153 | **0.2492** | 0.3521 | **0.3590** |
| CelebA | FedGAN [42] | 98.1784 | 0.3469 | 0.4210 | 0.1349 | 0.1929 |
| | FedDiffuse [63] | 33.3323 | 0.2986 | **0.5176** | **0.2318** | **0.2793** |
| | PRISM [67] | 200.1870 | 0.1479 | 0.1809 | 0.0684 | 0.0769 |
| | DfD | **29.1832** | **0.3734** | 0.4143 | 0.2229 | 0.2370 |

Table 2: Results on synthetic dataset utility.

| | | *LogReg* | *MLP* | *CNN* |
|---|---|---|---|---|
| MNIST | FedGAN [42] | 71.7 | 72.4 | 73.6 |
| | FedDiffuse [63] | 78.2 | 77.5 | 78.8 |
| | PRISM [67] | 43.1 | 41.4 | 45.3 |
| | DfD | **78.5** | **78.1** | **78.9** |
| CIFAR-10 | FedGAN [42] | 19.8 | 21.1 | 24.3 |
| | FedDiffuse [63] | **29.2** | 31.3 | 33.0 |
| | PRISM [67] | 11.5 | 12.9 | 13.2 |
| | DfD | 28.3 | **32.4** | **34.1** |
| CelebA | FedGAN [42] | 42.1 | 43.4 | 45.8 |
| | FedDiffuse [63] | 55.2 | **58.1** | 58.2 |
| | PRISM [67] | 12.2 | 11.3 | 13.4 |
| | DfD | **57.3** | 56.5 | **59.3** |

## 5.2 Results

**Quality and Utility.** Table 1 summarizes the quality-based results, i.e., FID, Precision (P), Recall (R), Density (D) and Coverage (D). Our method outperforms other FL methods for generative modeling in synthetic data fidelity. We provide generation results of each method in Figure 3. Table 2 summarizes the test accuracies as synthetic data utility. Following [84], we train three server-side classifiers on each generated synthetic dataset: logistic regression (*LogReg*), multi-layered perceptron (*MLP*), and convolution neural network (*CNN*). We evaluate each classifier on a separate test fold held in the central server. As a proxy of raw local data samples inaccessible in FL settings, synthetic dataset from DfD have been shown to offer better utility compared to existing baselines.

**Efficiency.** The communication costs differ in DfD compared to other methods. Table 3 summarizes the communication target and computation budget required to generate $N$ samples. During FL, DfD is faster in computation as it only conducts inferences on samples (i.e., $N$ forward passes for $N$ samples), whereas other methods require both backward and forward passes $N \times E$ times to update parameters. Additionally, DfD exchanges predictions of which size is $N \times d$, where the dimension is far smaller than the size of the model parameters (i.e., $d \ll p$. Thus, it can significantly reduce communication ($\because N \times d \ll p$) costs by setting reasonable number of samples, $N$.

Table 3: Comparison on communication cost & computation complexity.

| | Communication | Computation |
|---|---|---|
| `FedGAN` [42]
`FedDiffuse` [63]
`PRISM` [67] | $\boldsymbol{\theta}_i \in \mathbb{R}^p$ | $\mathcal{O}(N \times E \times p)$
(forward & backward $E$ times) |
| `DfD` | $\left\{ \boldsymbol{\epsilon}_{\boldsymbol{\theta}_i}(\boldsymbol{x}_t^{(j)}, t) \right\}_{j=1}^N \in \mathbb{R}^{N \times d}$ | $\mathcal{O}(N \times p)$
(single forward pass) |

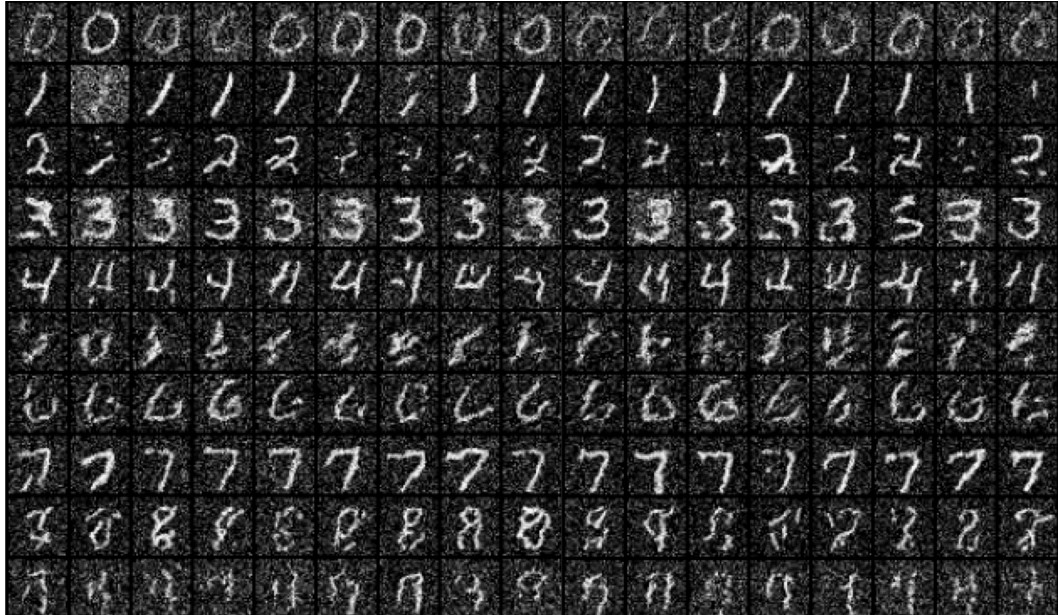

Figure 2: Differentially private synthetic dataset for MNIST from `DfD` under $(\epsilon = 10, \delta = 10^{-5})$-DP.

**Privacy.** Thanks to Theorem 4.7, `DfD` readily satisfies DP. Following [85], we let each client train its diffusion model with DP-SGD [73], under shard-partitioned non-IID setting [1] for $K = 10$ clients: each client has samples from only 2 out of 10 classes from MNIST dataset, i.e., digits 0 and 1 for client 0, digits 1 and to for client 1, ... , and digits 9 and 0 for client 9. To achieve $(\max_i \epsilon_i, \max_i \delta_i)$-DP for the resulting synthetic dataset, we set $\epsilon_i = 10$ and $\delta_i = 10^{-5}$ for all $i \in [K]$ clients. We found that applying DP is detrimental to sample quality as expected, and subtle tuning of step size is required to obtain discernible samples. Thereby, improving the quality of ULA sampling from differentially private diffusion models is a promising future direction for `DfD` in practice.

## 6 Limitation and Discussion

`DfD` gives clients great flexibility under the assumption of credible participation, such as cross-silo FL settings. In cross-device FL settings, where massive and unreliable clients [4] participate, `DfD` may fail, so we only consider cross-silo FL settings where a moderate number of credible clients participate. This could be relaxed by allowing partial participation through approximation of a global score at the central server [86].

Currently, the server ends up having synthetic dataset at last, not a generative model. Thus, by training a server-side amortized sampler [87–89] to emulate the collaborative sampling process, we can additionally generate samples even after the collaboration. Moreover, the communication cost can be further reduced by adapting advanced samplers [90–93] or by using model compression techniques, which we leave for future work.

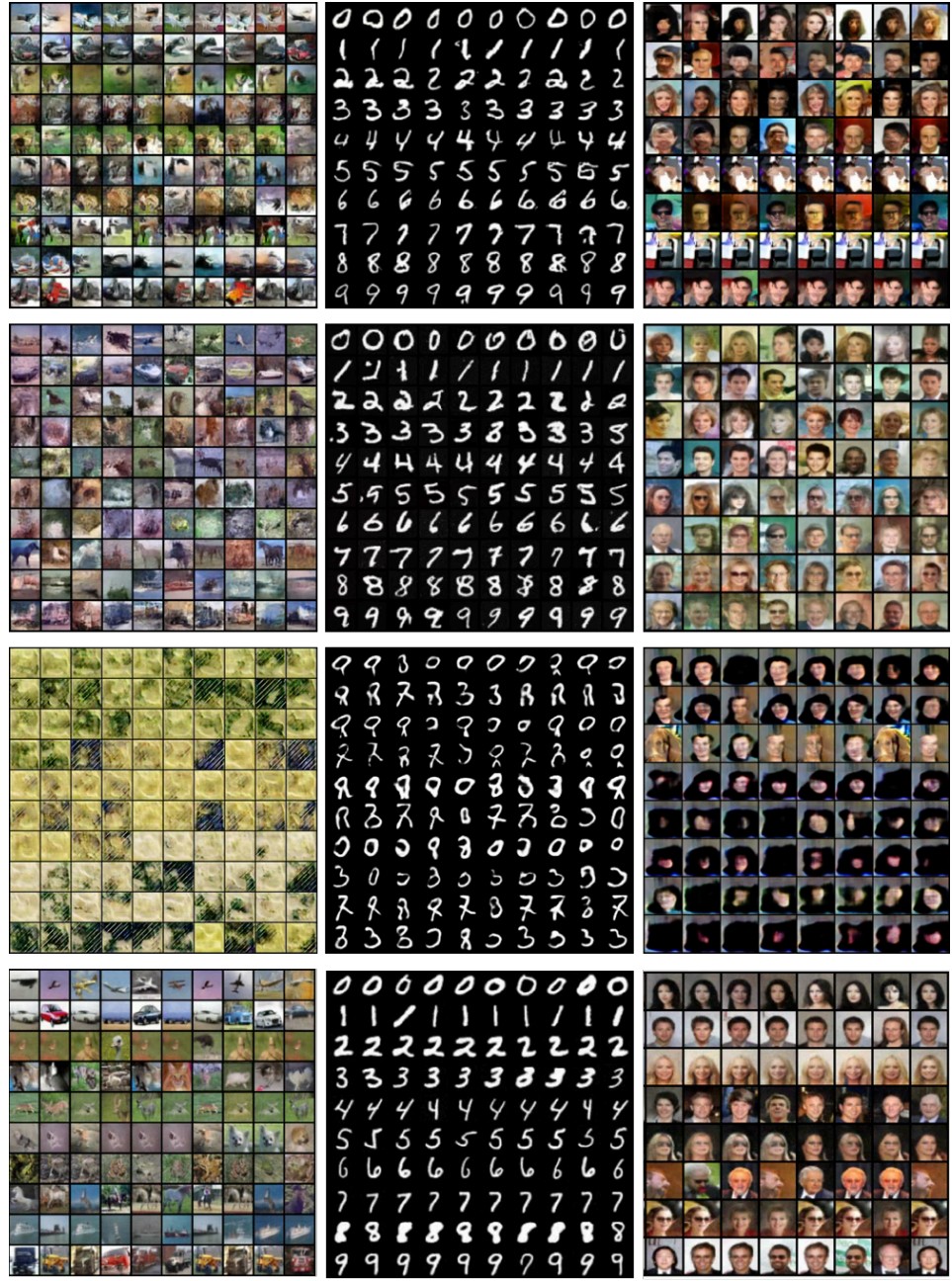

Figure 3: **Visualization of synthetic dataset generated under data-limited non-IID setting.** Each row corresponds to `FedGAN` [42], `FedDiffuse` [63], `PRSIM` [67], and `DfD`. Each column corresponds to CIFAR-10 [77], MNIST [76], and CelebA [78].

The success of `DfD` hinges on the faithful, authorized training of local diffusion models by participating clients. However, when local diffusion models are overfit or even memorize samples, this would introduce biased or collapsed sampling, resulting in catastrophic generation results. Therefore, careful training is required (e.g., earlystopping, weight decay) to acquire literally *well-trained* diffusion models. To guarantee trustworthy training, `DfD` requires a credible consortium of clients. Alternatively, we can use cryptographic tools, such as zero-knowledge proofs, to certify verified pre-training [94].

Lastly, thanks to the advancement in diffusion models, we expect `DfD` to be extended to other modalities than images [95, 96]. We believe the directions discussed thus far could improve the scalability and practicality of `DfD` in future works.

# 7 Conclusion

We propose a collaborative synthetic data generation framework, `DfD`, that leverages an energy-based connection for cooperative inference of diffusion models. `DfD` offers improvements in generation quality, communication efficiency, and easy privacy guarantees with theoretically grounded design. Given wide implications of synthetic data in federated settings, we look forward to exploring extensions of `DfD` to diverse modalities and data-intensive domains as a trustworthy framework.

## Broader Impact

The `DfD` framework enables federated synthetic data generation with privacy guarantees, promoting secure data sharing in privacy-sensitive domains. It produces high-quality synthetic data that preserves statistical properties, improving collaborative research and training of models while complying with e.g., GDPR [97] and HIPAA [98]. However, as synthetic datasets can be possibly misused for malicious purposes, a robust accounting protocol is required for ethical deployment.

## Acknowledgements

This research was conducted when Seok-Ju was at Seoul National University, supported by the National Research Foundation of Korea (NRF) Grant funded by the Korea Government under Grant No. RS-2025-00516776. The computational resource was supported by the High-Performance Computing Support Project, funded by the Government of the Republic of Korea (Ministry of Science and ICT) under Grant No. G2025-0146.

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

# Appendix

## Table of Contents

## A  Derivation of Gaussian Diffusion Models

Diffusion models are a class of generative models that aim to learn a data distribution $p_{\text{data}}(\boldsymbol{x}) \equiv q(\boldsymbol{x}_0)$ by learning to transform random Gaussian noise into original data through an iterative denoising process. In other words, the underlying Markov chain from the noise ($\boldsymbol{x}_T$) to the data ($\boldsymbol{x}_0$) defines diffusion models, and they are realized by two main processes: a forward process and a reverse process.

In the forward process, data $\boldsymbol{x}_0 \sim q(\boldsymbol{x}_0)$ is gradually perturbed over $T$ timesteps by adding Gaussian noise as

$$q(\boldsymbol{x}_t|\boldsymbol{x}_{t-1}) = \mathcal{N}(\boldsymbol{x}_t; \sqrt{1-\beta_t}\boldsymbol{x}_{t-1}, \beta_t\mathbf{I}_d),$$

where $\beta_t \in (0,1)$ controls the noise schedule, until $x_T \sim \mathcal{N}(\mathbf{0}_d, \mathbf{I}_d)$. Other constants satisfy $\alpha_t = 1 - \beta_t$ and $\bar{\alpha}_t = \prod_{\tau=1}^T \alpha_\tau$. Thus, the forward process models $q(\boldsymbol{x}_1, ..., \boldsymbol{x}_T|\boldsymbol{x}_0) = \prod_{t=1}^T q(\boldsymbol{x}_t|\boldsymbol{x}_{t-1})$.

To reiterate, as in Eq. (2), diffusion models have useful property that enables calculation of anytime marginal distribution in a closed form:

$$q(\boldsymbol{x}_t|\boldsymbol{x}_0) = \mathcal{N}(\boldsymbol{x}_t; \sqrt{\bar{\alpha}_t}\boldsymbol{x}_0, (1-\bar{\alpha}_t)\mathbf{I}_d)$$

By training a parameterized deep network, diffusion models can denoise from noise to the data by approximating the true posterior $p_{\boldsymbol{\theta}}(\boldsymbol{x}_{t-1}|\boldsymbol{x}_t) \approx q(\boldsymbol{x}_{t-1}|\boldsymbol{x}_t, \boldsymbol{x}_0)$, through the reverse process as defined in Eq. (1). Thus, the reverse process models $p_{\boldsymbol{\theta}}(\boldsymbol{x}_0, ..., \boldsymbol{x}_T) = p(\boldsymbol{x}_T) \prod_{t=1}^T p_{\boldsymbol{\theta}}(\boldsymbol{x}_{t-1}|\boldsymbol{x}_t)$.

With these two paired processes, diffusion models maximize the lower bound of log-likelihood defined as:

$$
\begin{aligned}
\log p_{\boldsymbol{\theta}}(\boldsymbol{x}_0) &\geq \mathbb{E}_{q(\boldsymbol{x}_1, ..., \boldsymbol{x}_T|\boldsymbol{x}_0)}\left[\log \frac{p_{\boldsymbol{\theta}}(\boldsymbol{x}_0, ..., \boldsymbol{x}_T)}{q(\boldsymbol{x}_1, ..., \boldsymbol{x}_T|\boldsymbol{x}_0)}\right] \\
&= \log p_{\boldsymbol{\theta}}(\boldsymbol{x}_0) - \mathrm{D}_{\text{KL}}(q(\boldsymbol{x}_1, ..., \boldsymbol{x}_T|\boldsymbol{x}_0) \parallel p_{\boldsymbol{\theta}}(\boldsymbol{x}_1, ..., \boldsymbol{x}_T|\boldsymbol{x}_0)) \\
&= \log p_{\boldsymbol{\theta}}(\boldsymbol{x}_0) - \sum_{t=1}^T \mathrm{D}_{\text{KL}}(q(\boldsymbol{x}_{t-1}|\boldsymbol{x}_t, \boldsymbol{x}_0) \parallel p_{\boldsymbol{\theta}}(\boldsymbol{x}_{t-1}|\boldsymbol{x}_t)),
\end{aligned}
$$

where the decomposition is due to the Markov property of both forward and reverse processes.

From this, we can maximize the lower bound of log-likelihood by minimizing the sum of KL divergence terms instead:

$$\sum_{t=1}^T \mathrm{D}_{\text{KL}}(q(\boldsymbol{x}_{t-1}|\boldsymbol{x}_t, \boldsymbol{x}_0) \parallel p_{\boldsymbol{\theta}}(\boldsymbol{x}_{t-1}|\boldsymbol{x}_t)).$$

Since both $q(\boldsymbol{x}_{t-1}|\boldsymbol{x}_t, \boldsymbol{x}_0)$ and $p_{\boldsymbol{\theta}}(\boldsymbol{x}_{t-1}|\boldsymbol{x}_t)$ are Gaussian, the KL divergence simplifies to a mean squared error between the true noise $\boldsymbol{\epsilon}$ and the estimated noise $\boldsymbol{\epsilon}_{\boldsymbol{\theta}}(\boldsymbol{x}_t, t)$. Hence, we have

$$\sum_{t=1}^{T} D_{\mathrm{KL}}(q(\boldsymbol{x}_{t-1}|\boldsymbol{x}_t, \boldsymbol{x}_0) \parallel p_{\boldsymbol{\theta}}(\boldsymbol{x}_{t-1}|\boldsymbol{x}_t))$$

$$= \sum_{t=1}^{T} a_t \mathbb{E}_{\boldsymbol{x}_0 \sim q(\boldsymbol{x}_0), \boldsymbol{\epsilon} \sim \mathcal{N}(\mathbf{0}_d, \mathbf{I}_d)} \left[ \|\boldsymbol{\epsilon} - \boldsymbol{\epsilon}_{\boldsymbol{\theta}}(\boldsymbol{x}_t, t)\|_2^2 \right]$$

$$=: \sum_{t=1}^{T} a_t \mathcal{L}(\boldsymbol{\theta}, t),$$

where $a_t$ is a weight that is typically treated equal as $a_1 = ... = a_T = 1$ [9] for all time-dependent loss $\mathcal{L}(\boldsymbol{\theta}, t)$, which was defined in Eq. (3).

After the training is completed by optimizing the above composite loss $\mathcal{L}(\boldsymbol{\theta}) = \sum_{t=1}^{T} \mathcal{L}(\boldsymbol{\theta}, t)$, we can draw samples through ancestral sampling: starting from $\boldsymbol{x}_T \sim \mathcal{N}(\mathbf{0}_d, \mathbf{I}_d)$ using $\boldsymbol{\mu}_{\boldsymbol{\theta}}(\boldsymbol{x}_t, t)$, due to the connection $\boldsymbol{x}_{t-1} = \boldsymbol{\mu}_{\boldsymbol{\theta}}(\boldsymbol{x}_t, t) + \sqrt{\tilde{\beta}_t} \boldsymbol{\epsilon}, \quad \boldsymbol{\epsilon} \sim \mathcal{N}(\mathbf{0}_d, \mathbf{I}_d)$ from the reverse process. Note here that $\boldsymbol{\mu}_{\boldsymbol{\theta}}(\boldsymbol{x}_t, t)$ is computed from the estimated noise $\boldsymbol{\epsilon}_{\boldsymbol{\theta}}(\boldsymbol{x}_t, t)$.

## B  Connection of Energy Based Models and Diffusion Models

EBMs and diffusion models share a profound theoretical connection through denoising score matching in Eq. (5). This connection not only provides an alternative interpretation of diffusion models but also enables ULA in Eq. (8).

Again, EBMs model an unnormalized probability density of the form of:

$$p_{\boldsymbol{\theta}}(\boldsymbol{x}) = \frac{\exp(-\lambda f_{\boldsymbol{\theta}}(\boldsymbol{x}))}{Z_{\boldsymbol{\theta}}},$$

where $f_{\boldsymbol{\theta}} : \mathbb{R}^d \to \mathbb{R}$ is the energy function parameterized by $\boldsymbol{\theta} \in \mathbb{R}^p$, $\lambda \in \mathbb{R}^+$ is a scale factor, and $Z_{\boldsymbol{\theta}} = \int_{\boldsymbol{x} \in \mathcal{X}} \exp(-\lambda f_{\boldsymbol{\theta}}(\boldsymbol{x})) \, \mathrm{d}\boldsymbol{x}$ is the normalizing constant, which is typically intractable in practice.

Due to the intractable property of the normalizing constant $Z_{\boldsymbol{\theta}}$, direct computation of the likelihood is challenging. This necessitates alternative training methods using a score, defined as follows. The *score* is the gradient of the log-density as:

$$\nabla_{\boldsymbol{x}} \log p_{\boldsymbol{\theta}}(\boldsymbol{x}) = -\lambda \nabla_{\boldsymbol{x}} f_{\boldsymbol{\theta}}(\boldsymbol{x}).$$

To train EBMs, we use denoising score matching objective [69] in Eq. (5) to minimize the Fisher divergence between the score of a model's distribution and the score of a noise-perturbed data distribution:

$$q(\boldsymbol{x}_\sigma) = \int_{\boldsymbol{x} \in \mathcal{X}} q_\sigma(\boldsymbol{x}_\sigma | \boldsymbol{x}) p_{\mathrm{data}}(\boldsymbol{x}) \, \mathrm{d}\boldsymbol{x},$$

where the perturbation is realized as:

$$\boldsymbol{x}_\sigma = \boldsymbol{x} + \sigma \boldsymbol{\epsilon}, \quad \boldsymbol{\epsilon} \sim \mathcal{N}(\mathbf{0}_d, \mathbf{I}_d),$$

and $\sigma$ is the noise scale. Hence, the DSM objective is:

$$\mathcal{L}(\boldsymbol{\theta}, \sigma) = \mathbb{E}_{\boldsymbol{x}_\sigma \sim q(\boldsymbol{x}_\sigma | \boldsymbol{x}), \boldsymbol{x} \sim p_{\mathrm{data}}(\boldsymbol{x})} \left[ \|\nabla_{\boldsymbol{x}_\sigma} \log q(\boldsymbol{x}_\sigma | \boldsymbol{x}) - \nabla_{\boldsymbol{x}_\sigma} \log p_{\boldsymbol{\theta}}(\boldsymbol{x}_\sigma) \|_2^2 \right]$$

Rewriting $q(\boldsymbol{x}_\sigma | \boldsymbol{x}) = \mathcal{N}(\boldsymbol{x}_\sigma; \boldsymbol{x}, \sigma^2 \mathbf{I}_d)$, we can explicitly have that

$$\nabla_{\boldsymbol{x}_\sigma} \log q(\boldsymbol{x}_\sigma | \boldsymbol{x}) = -\frac{(\boldsymbol{x}_\sigma - \boldsymbol{x})}{\sigma^2} = -\sigma \boldsymbol{\epsilon}.$$

By substituting the EBM score as $\nabla_{\boldsymbol{x}_\sigma} \log p_{\boldsymbol{\theta}}(\boldsymbol{x}_\sigma) = -\lambda \nabla_{\boldsymbol{x}_\sigma} f_{\boldsymbol{\theta}}(\boldsymbol{x}_\sigma)$, the objective becomes equivalent (up to a constant) to Eq. (6) as:

$$\sigma^2 \mathcal{L}(\boldsymbol{\theta}, \sigma) = \mathbb{E}_{\boldsymbol{x}_\sigma \sim q(\boldsymbol{x}_\sigma | \boldsymbol{x}), \boldsymbol{\epsilon} \sim \mathcal{N}(\mathbf{0}_d, \mathbf{I}_d)} \left[ \|\boldsymbol{\epsilon} - \sigma \lambda \nabla_{\boldsymbol{x}_\sigma} f_{\boldsymbol{\theta}}(\boldsymbol{x}_\sigma) \|_2^2 \right].$$

Diffusion models, as defined in Section A, optimize a similar objective. To reiterate, the objective of diffusion models is given as:

$$\mathcal{L}(\boldsymbol{\theta}, t) = \mathbb{E}_{\boldsymbol{x}_0, \boldsymbol{\epsilon}} \left[ \|\boldsymbol{\epsilon} - \boldsymbol{\epsilon}_{\boldsymbol{\theta}}(\boldsymbol{x}_t, t) \|_2^2 \right].$$

From this, we can easily draw an analogy with DSM objective, by replacing the $\sigma$ with a time-dependent noise scale $\sigma_t$, with the score interpretation as:

$$\nabla_{\boldsymbol{x}_{\sigma_t}} \log p_{\boldsymbol{\theta}}(\boldsymbol{x}_{\sigma_t}) = -\lambda \nabla_{\boldsymbol{x}_{\sigma_t}} f_{\boldsymbol{\theta}}(\boldsymbol{x}_{\sigma_t}) \equiv -\frac{\boldsymbol{\epsilon}_{\boldsymbol{\theta}}(\boldsymbol{x}_t, t)}{\sigma_t}.$$

This connection shows that the noise prediction $\boldsymbol{\epsilon}_{\boldsymbol{\theta}}(\boldsymbol{x}_t, t)$ in diffusion models directly corresponds to the score of an implicit EBM, if scaled by the noise level $\sigma_t$. Thus, diffusion models can be viewed as learning EBMs implicitly where the score is approximated by the noise prediction network parameterized by $\boldsymbol{\theta}$. This energy-based interpretation allows for alternative sampling methods in diffusion models, such as ULA in Eq. (8). Note that for the compositional generation, this requires the energy-based parameterization tricks of diffusion models, discussed in Section 3.3 and Section 3.4.

# C  Proofs

## C.1  Proof of Lemma 4.3

*Proof.* In this proof, we need to show $\mathbb{E}\left[\|\boldsymbol{x}_{t-1}\|_2^2 - \|\boldsymbol{x}_t\|_2^2\right] \leq 0$ from diffusion reverse process in Eq. (1). First, following [9], we equivalently define for variance schedule constants $\beta_t, t \in [T]$ that other constants are defined as follows.

$$\alpha_t = 1 - \beta_t, \quad \bar{\alpha}_t = \prod_{\tau=1}^{t} \alpha_\tau, \quad \tilde{\beta}_t = \beta_t \frac{1 - \bar{\alpha}_{t-1}}{1 - \bar{\alpha}_t}.$$

Recall that the reverse process in Eq. (1) can be written as

$$\boldsymbol{x}_{t-1} = \boldsymbol{\mu}_\theta(\boldsymbol{x}_t, t) + \tilde{\beta}_t \boldsymbol{z} = \frac{1}{\sqrt{\alpha_t}}\left(\boldsymbol{x}_t - \frac{\beta_t}{\sqrt{1 - \bar{\alpha}_t}}\boldsymbol{\epsilon}_\theta(\boldsymbol{x}_t, t)\right) + \tilde{\beta}_t \boldsymbol{z}, \quad \boldsymbol{z} \sim \mathcal{N}(\boldsymbol{0}_d, \mathbf{I}_d).$$

From this, we have from the law of total expectation that

$$\mathbb{E}\left[\|\boldsymbol{x}_{t-1}\|_2^2\right] = \mathbb{E}\left[\|\boldsymbol{\mu}_\theta(\boldsymbol{x}_t, t)\|_2^2\right] + \tilde{\beta}_t d$$

For $\mathbb{E}\left[\|\boldsymbol{\mu}_\theta(\boldsymbol{x}_t, t)\|_2^2\right]$, we have

$$
\begin{aligned}
&\mathbb{E}\left[\|\boldsymbol{\mu}_\theta(\boldsymbol{x}_t, t)\|_2^2\right] \\
&= \frac{1}{\alpha_t}\left(\mathbb{E}\left[\|\boldsymbol{x}_t\|_2^2\right] - \frac{2\beta_t}{\sqrt{1 - \bar{\alpha}_t}}\mathbb{E}\left[\langle\boldsymbol{x}_t, \boldsymbol{\epsilon}_\theta(\boldsymbol{x}_t, t)\rangle\right] + \frac{\beta_t^2}{1 - \bar{\alpha}_t}\mathbb{E}\left[\|\boldsymbol{\epsilon}_\theta(\boldsymbol{x}_t, t)\|_2^2\right]\right) \\
&\underset{\text{Definition 4.1}}{\approx} \frac{1}{\alpha_t}\left(\mathbb{E}\left[\|\boldsymbol{x}_t\|_2^2\right] - \frac{2\beta_t}{\sqrt{1 - \bar{\alpha}_t}}\mathbb{E}\left[\langle\boldsymbol{x}_t, \boldsymbol{\epsilon}\rangle\right] + \frac{\beta_t^2}{1 - \bar{\alpha}_t}\mathbb{E}\left[\|\boldsymbol{\epsilon}\|_2^2\right]\right) \\
&= \frac{1}{\alpha_t}\left(\mathbb{E}\left[\|\boldsymbol{x}_t\|_2^2\right] - \frac{2\beta_t}{\sqrt{1 - \bar{\alpha}_t}} \cdot \sqrt{1 - \bar{\alpha}_t}d + \frac{\beta_t^2}{1 - \bar{\alpha}_t}d\right) \\
&= \frac{1}{\alpha_t}\left(\mathbb{E}\left[\|\boldsymbol{x}_t\|_2^2\right] - 2\beta_t d + \frac{\beta_t^2}{1 - \bar{\alpha}_t}d\right),
\end{aligned}
$$

where we used $\mathbb{E}\left[\|\boldsymbol{\epsilon}\|_2^2\right] = d$ for any $\boldsymbol{\epsilon} \sim \mathcal{N}(\boldsymbol{0}_d, \mathbf{I}_d)$ and $\mathbb{E}\left[\langle\boldsymbol{x}_t, \boldsymbol{\epsilon}\rangle\right] = \sqrt{1 - \bar{\alpha}_t}d$ from Eq. (2). Thus, we have

$$\mathbb{E}\left[\|\boldsymbol{x}_{t-1}\|_2^2 - \|\boldsymbol{x}_t\|_2^2\right] = \left(\frac{1}{\alpha_t} - 1\right)\mathbb{E}\left[\|\boldsymbol{x}_t\|_2^2\right] - \frac{2\beta_t d}{\alpha_t} + \frac{\beta_t^2 d}{\alpha_t(1 - \bar{\alpha}_t)} + \tilde{\beta}_t d$$

From Eq. (2), we have

$$\mathbb{E}[\|\boldsymbol{x}_t\|^2] = \bar{\alpha}_t\mathbb{E}\left[\|\boldsymbol{x}_0\|_2^2\right] + (1 - \bar{\alpha}_t)d,$$

as $\boldsymbol{x}_0$ and $\boldsymbol{\epsilon}$ are independent and $\mathbb{E}\left[\|\boldsymbol{\epsilon}\|_2^2\right] = d$.

Using this, we derive

$$
\mathbb{E}\left[\|\boldsymbol{x}_{t-1}\|_2^2 - \|\boldsymbol{x}_t\|_2^2\right]
$$

$$
= \left(\frac{1}{\alpha_t} - 1\right)\left(\bar{\alpha}_t \mathbb{E}\left[\|\boldsymbol{x}_0\|_2^2\right] + (1-\bar{\alpha}_t)d\right) - \frac{2\beta_t d}{\alpha_t} + \frac{\beta_t^2 d}{\alpha_t(1-\bar{\alpha}_t)} + \tilde{\beta}_t d
$$

$$
\leq \left(\frac{1}{\alpha_t} - 1 - \frac{2\beta_t}{\alpha_t} + \frac{\beta_t^2}{\alpha_t(1-\bar{\alpha}_t)} + \tilde{\beta}_t\right)d
$$

$$
= \left(\frac{1}{\alpha_t} - 1 - \frac{2\beta_t}{\alpha_t} + \frac{\beta_t^2}{\alpha_t(1-\bar{\alpha}_t)} + \frac{1-\bar{\alpha}_{t-1}}{1-\bar{\alpha}_t}\beta_t\right)d
$$

$$
= \left(\frac{1-\alpha_t}{\alpha_t} - \frac{2\beta_t}{\alpha_t} + \frac{\beta_t^2}{\alpha_t(1-\bar{\alpha}_t)} + \frac{1-\bar{\alpha}_{t-1}}{1-\bar{\alpha}_t}\beta_t\right)d
$$

$$
= \left(\frac{\beta_t}{\alpha_t} - \frac{2\beta_t}{\alpha_t} + \frac{\beta_t^2}{\alpha_t(1-\bar{\alpha}_t)} + \frac{1-\bar{\alpha}_{t-1}}{1-\bar{\alpha}_t}\beta_t\right)d
$$

$$
= \left(-\frac{\beta_t}{\alpha_t} + \frac{\beta_t^2}{\alpha_t(1-\bar{\alpha}_t)} + \frac{1-\bar{\alpha}_{t-1}}{1-\bar{\alpha}_t}\beta_t\right)d
$$

$$
= \frac{\beta_t}{\alpha_t(1-\bar{\alpha}_t)}\left(-(1-\bar{\alpha}_t) + \beta_t + \alpha_t(1-\bar{\alpha}_{t-1})\right),
$$

where the first inequality is due to $\|\boldsymbol{x}\|_2 \leq \sqrt{d}\|\boldsymbol{x}\|_\infty, \forall \boldsymbol{x} \in \mathbb{R}^d$, along with typical assumption in diffusion models that $\|\boldsymbol{x}_0\|_\infty = 1$ as inputs are normalized into $[-1, 1]^d$ [9].

Rearranging, we have

$$
\mathbb{E}\left[\|\boldsymbol{x}_{t-1}\|_2^2 - \|\boldsymbol{x}_t\|_2^2\right]
$$

$$
\leq \frac{\beta_t}{\alpha_t(1-\bar{\alpha}_t)}\left(-(1-\bar{\alpha}_t) + \beta_t + \alpha_t(1-\bar{\alpha}_{t-1})\right)
$$

$$
= \frac{\beta_t}{\alpha_t(1-\bar{\alpha}_t)}\left(-(1-\beta_t) + \alpha_t + \bar{\alpha}_t - \alpha\bar{\alpha}_{t-1})\right)
$$

$$
= \frac{\beta_t}{\alpha_t(1-\bar{\alpha}_t)}\left(-\alpha_t + \alpha_t + \bar{\alpha}_t - \alpha\bar{\alpha}_{t-1})\right)
$$

$$
= \frac{\beta_t}{\alpha_t(1-\bar{\alpha}_t)}\left(-\alpha_t + \alpha_t + \bar{\alpha}_t - \bar{\alpha}_t\right)
$$

$$
= 0,
$$

where the second last and the third last equalities are due to the definition of $\bar{\alpha}_t$ and $\beta_t$ each.

We finally have $\mathbb{E}\left[\|\boldsymbol{x}_{t-1}\|_2^2 - \|\boldsymbol{x}_t\|_2^2\right] \leq 0$, thus $\mathbb{E}\left[\|\boldsymbol{x}_{t-1}\|_2^2\right] \leq \mathbb{E}\left[\|\boldsymbol{x}_t\|_2^2\right]$. $\qquad\square$

### C.2 Proof of Lemma 4.4

#### C.2.1 Proofs

*Proof.* In this proof, we need to show $\mathbb{E}\left[\|\boldsymbol{x}_{t-1}\|_2^2 - \|\boldsymbol{x}_t\|_2^2\right] \leq 0$ from ULA update in Eq. (8).

With the energy-based $\ell_2$ parameterization in Eq. (11), denote from ULA update that

$$
\Delta\boldsymbol{x}_t := \boldsymbol{x}_{t-1} - \boldsymbol{x}_t = -\frac{\lambda\eta_t}{\sqrt{1-\bar{\alpha}_t}}\boldsymbol{\epsilon}_{\boldsymbol{\theta}}(\boldsymbol{x}_t, t) + \sqrt{2\eta_t}\boldsymbol{z}_t, \quad \boldsymbol{z}_t \sim \mathcal{N}(\mathbf{0}_d, \mathbf{I}_d).
$$

With this, we have that

$$
\|\boldsymbol{x}_{t-1}\|_2^2 - \|\boldsymbol{x}_t\|_2^2 = \|\boldsymbol{x}_t + \Delta\boldsymbol{x}_t\|_2^2 - \|\boldsymbol{x}_t\|_2^2 = 2\langle x_t, \Delta\boldsymbol{x}_t\rangle + \|\Delta\boldsymbol{x}_t\|_2^2.
$$

Now, taking expectations over $\boldsymbol{z}_t$ and $\boldsymbol{x}_t|\boldsymbol{x}_0$, we have that

$$
\mathbb{E}_{\boldsymbol{x}_t|\boldsymbol{x}_0}\left[\mathbb{E}_{\boldsymbol{z}_t}\left[\|\boldsymbol{x}_{t-1}\|_2^2 - \|\boldsymbol{x}_t\|_2^2\right]\right]
$$

$$
= \mathbb{E}_{\boldsymbol{x}_t|\boldsymbol{x}_0}\left[\mathbb{E}_{\boldsymbol{z}_t}\left[2\langle\boldsymbol{x}_t, \Delta\boldsymbol{x}_t\rangle + \|\Delta\boldsymbol{x}_t\|_2^2\right]\right] \qquad\qquad \text{(A1)}
$$

$$
= 2\mathbb{E}_{\boldsymbol{x}_t|\boldsymbol{x}_0}\left[\langle\boldsymbol{x}_t, \mathbb{E}_{\boldsymbol{z}_t}[\Delta\boldsymbol{x}_t]\rangle\right] + \mathbb{E}_{\boldsymbol{x}_t|\boldsymbol{x}_0}\left[\mathbb{E}_{\boldsymbol{z}_t}[\|\Delta\boldsymbol{x}_t\|_2^2]\right].
$$

Let us demystify the inner expectation first. Since $\mathbb{E}_{\boldsymbol{z}_t}[\boldsymbol{z}_t] = 0$, we have that

$$\mathbb{E}_{\boldsymbol{z}_t}[\Delta \boldsymbol{x}_t] = -\frac{\lambda \eta_t}{\sqrt{1 - \bar{\alpha}_t}} \boldsymbol{\epsilon}_{\boldsymbol{\theta}}(\boldsymbol{x}_t, t).$$

Next, for $\mathbb{E}_{\boldsymbol{z}_t}[\|\Delta \boldsymbol{x}_t\|_2^2]$, we have that

$$\mathbb{E}_{\boldsymbol{z}_t}\left[\frac{\lambda^2 \eta_t^2}{1 - \bar{\alpha}_t}\|\boldsymbol{\epsilon}_{\boldsymbol{\theta}}(\boldsymbol{x}_t, t)\|_2^2 - \frac{2\lambda \eta_t \sqrt{2\eta_t}}{\sqrt{1 - \bar{\alpha}_t}}\langle \boldsymbol{\epsilon}_{\boldsymbol{\theta}}(\boldsymbol{x}_t, t), \boldsymbol{z}_t\rangle + 2\eta_t \|\boldsymbol{z}_t\|^2\right]$$

$$= \frac{\lambda^2 \eta_t^2}{1 - \bar{\alpha}_t}\|\boldsymbol{\epsilon}_{\boldsymbol{\theta}}(\boldsymbol{x}_t, t)\|_2^2 - \frac{2\lambda \eta_t \sqrt{2\eta_t}}{\sqrt{1 - \bar{\alpha}_t}}\langle \boldsymbol{\epsilon}_{\boldsymbol{\theta}}(\boldsymbol{x}_t, t), \mathbb{E}_{\boldsymbol{z}_t}[\boldsymbol{z}_t]\rangle + 2\eta_t \mathbb{E}_{\boldsymbol{z}_t}[\|\boldsymbol{z}_t\|_2^2]$$

$$= \frac{\lambda^2 \eta_t^2}{1 - \bar{\alpha}_t}\|\boldsymbol{\epsilon}_{\boldsymbol{\theta}}(\boldsymbol{x}_t, t)\|_2^2 + 2\eta_t d,$$

where $\mathbb{E}[\|\boldsymbol{z}_t\|_2^2] = d$ for $\boldsymbol{z}_t \sim \mathcal{N}(\boldsymbol{0}_d, \mathbf{I}_d)$.

To sum up, for the inner expectation of Eq. (A1), we have that

$$\mathbb{E}_{\boldsymbol{z}_t}\left[\|\boldsymbol{x}_{t-1}\|_2^2 - \|\boldsymbol{x}_t\|_2^2\right] = -\frac{2\lambda \eta_t}{\sqrt{1 - \bar{\alpha}_t}}\langle \boldsymbol{x}_t, \boldsymbol{\epsilon}_{\boldsymbol{\theta}}(\boldsymbol{x}_t, t)\rangle + \frac{\lambda^2 \eta_t^2}{1 - \bar{\alpha}_t}\|\boldsymbol{\epsilon}_{\boldsymbol{\theta}}(\boldsymbol{x}_t, t)\|_2^2 + 2\eta_t d.$$

Going on for the outer expectation, we have that

$$\mathbb{E}_{\boldsymbol{x}_t|\boldsymbol{x}_0}\left[\mathbb{E}_{\boldsymbol{z}_t}\left[\|\boldsymbol{x}_{t-1}\|_2^2 - \|\boldsymbol{x}_t\|_2^2\right]\right]$$

$$= -\frac{2\lambda \eta_t}{\sqrt{1 - \bar{\alpha}_t}}\mathbb{E}_{\boldsymbol{x}_t|\boldsymbol{x}_0}\left[\langle \boldsymbol{x}_t, \boldsymbol{\epsilon}_{\boldsymbol{\theta}}(\boldsymbol{x}_t, t)\rangle\right] + \frac{\lambda^2 \eta_t^2}{1 - \bar{\alpha}_t}\mathbb{E}_{\boldsymbol{x}_t|\boldsymbol{x}_0}\left[\|\boldsymbol{\epsilon}_{\boldsymbol{\theta}}(\boldsymbol{x}_t, t)\|_2^2\right] + 2\eta_t d.$$

Since it is for well-trained diffusion models, we have by using Eq. (2) that

$$\mathbb{E}_{\boldsymbol{x}_t|\boldsymbol{x}_0}\left[\mathbb{E}_{\boldsymbol{z}_t}\left[\|\boldsymbol{x}_{t-1}\|_2^2 - \|\boldsymbol{x}_t\|_2^2\right]\right]$$

$$= -\frac{2\lambda \eta_t}{\sqrt{1 - \bar{\alpha}_t}}\mathbb{E}_{\boldsymbol{x}_t|\boldsymbol{x}_0}\left[\left\langle \boldsymbol{x}_t, \frac{\boldsymbol{x}_t - \sqrt{\bar{\alpha}_t}\boldsymbol{x}_0}{\sqrt{1 - \bar{\alpha}_t}}\right\rangle\right] \qquad (A2)$$

$$+ \frac{\lambda^2 \eta_t^2}{1 - \bar{\alpha}_t}\mathbb{E}_{\boldsymbol{x}_t|\boldsymbol{x}_0}\left[\left\|\frac{\boldsymbol{x}_t - \sqrt{\bar{\alpha}_t}\boldsymbol{x}_0}{\sqrt{1 - \bar{\alpha}_t}}\right\|_2^2\right] + 2\eta_t d.$$

From this, the first conditional expectation becomes that

$$\mathbb{E}_{\boldsymbol{x}_t|\boldsymbol{x}_0}\left[\left\langle \boldsymbol{x}_t, \frac{\boldsymbol{x}_t - \sqrt{\bar{\alpha}_t}\boldsymbol{x}_0}{\sqrt{1 - \bar{\alpha}_t}}\right\rangle\right]$$

$$= \frac{1}{\sqrt{1 - \bar{\alpha}_t}}\mathbb{E}_{\boldsymbol{x}_t|\boldsymbol{x}_0}\left[\langle \boldsymbol{x}_t, \boldsymbol{x}_t - \sqrt{\bar{\alpha}_t}\boldsymbol{x}_0\rangle\right] = \frac{1}{\sqrt{1 - \bar{\alpha}_t}}\mathbb{E}_{\boldsymbol{x}_t|\boldsymbol{x}_0}\left[\|\boldsymbol{x}_t\|_2^2 - \sqrt{\bar{\alpha}_t}\langle \boldsymbol{x}_t, \boldsymbol{x}_0\rangle\right],$$

where the former term inside the expectation is that

$$\mathbb{E}_{\boldsymbol{x}_t|\boldsymbol{x}_0}\left[\|\boldsymbol{x}_t\|_2^2\right] = \mathbb{E}_{\boldsymbol{x}_t|\boldsymbol{x}_0}\left[\sum_{i=1}^d (\boldsymbol{x}_i)^2\right] = \sum_{i=1}^d \mathbb{E}_{\boldsymbol{x}_t|\boldsymbol{x}_0}\left[(\boldsymbol{x}_i)^2\right]$$

$$= \sum_{i=1}^d \mathbb{E}_{\boldsymbol{x}_t|\boldsymbol{x}_0}\left[\mathrm{Var}[\boldsymbol{x}_{t,i}|\boldsymbol{x}_0] + (\mathbb{E}[\boldsymbol{x}_{t,i}|\boldsymbol{x}_0])^2\right]$$

$$= \sum_{i=1}^d \left\{(1 - \bar{\alpha}_t) + (\sqrt{\bar{\alpha}_t}\boldsymbol{x}_{0,i})^2\right\} = (1 - \bar{\alpha}_t)d + \bar{\alpha}_t \|\boldsymbol{x}_0\|_2^2,$$

and the second term inside the expectation is that

$$\mathbb{E}_{\boldsymbol{x}_t|\boldsymbol{x}_0}\left[\langle \boldsymbol{x}_t, \boldsymbol{x}_0\rangle\right] = \mathbb{E}_{\boldsymbol{x}_t|\boldsymbol{x}_0}\left[\sum_{i=1}^d \boldsymbol{x}_{0,i}\boldsymbol{x}_{t,i}\right] = \sum_{i=1}^d \boldsymbol{x}_{0,i}\mathbb{E}_{\boldsymbol{x}_t|\boldsymbol{x}_0}\left[\boldsymbol{x}_{t,i}\right]$$

$$= \sum_{i=1}^d \boldsymbol{x}_{0,i}\left(\sqrt{\bar{\alpha}_t}\boldsymbol{x}_{0,i}\right) = \sqrt{\bar{\alpha}_t}\sum_{i=1}^d \boldsymbol{x}_{0,i}^2 = \sqrt{\bar{\alpha}_t}\|\boldsymbol{x}_0\|_2^2.$$

Taken together, we have for the first conditional expectation that

$$\mathbb{E}_{\boldsymbol{x}_t|\boldsymbol{x}_0}\left[\left\langle \boldsymbol{x}_t, \frac{\boldsymbol{x}_t - \sqrt{\bar{\alpha}_t}\boldsymbol{x}_0}{\sqrt{1-\bar{\alpha}_t}} \right\rangle\right] = \frac{1}{\sqrt{1-\bar{\alpha}_t}}\left((1-\bar{\alpha}_t)d + \bar{\alpha}_t\|\boldsymbol{x}_0\|_2^2 - \sqrt{\bar{\alpha}_t}\cdot\sqrt{\bar{\alpha}_t}\|\boldsymbol{x}_0\|_2^2\right) = \sqrt{1-\bar{\alpha}_t}d.$$

Next, for the second conditional expectation term in Eq. (A2), we have that

$$\mathbb{E}_{\boldsymbol{x}_t|\boldsymbol{x}_0}\left[\left\|\frac{\boldsymbol{x}_t - \sqrt{\bar{\alpha}_t}\boldsymbol{x}_0}{\sqrt{1-\bar{\alpha}_t}}\right\|_2^2\right] = \mathbb{E}_{\boldsymbol{x}_t|\boldsymbol{x}_0}\left[\|\boldsymbol{\epsilon}\|_2^2\right] = d,$$

due to Eq. (2) and it is for well-trained diffusion models.

Putting all together, the original expectation in Eq. (A2) becomes that

$$\mathbb{E}_{\boldsymbol{x}_t|\boldsymbol{x}_0}\left[\mathbb{E}_{\boldsymbol{z}_t}\left[\|\boldsymbol{x}_{t-1}\|_2^2 - \|\boldsymbol{x}_t\|_2^2\right]\right]$$

$$= -\frac{2\lambda\eta_t}{\sqrt{1-\bar{\alpha}_t}}\cdot\sqrt{1-\bar{\alpha}_t}d + \frac{\lambda^2\eta_t^2}{1-\bar{\alpha}_t}\cdot d + 2\eta_t d$$

$$= \left(-2\lambda + \frac{\lambda^2\eta_t}{1-\bar{\alpha}_t} + 2\right)\eta_t d.$$

Since we want to guarantee this term to be non-increasing for the non-expansiveness w.r.t. L2 norm as in Lemma 4.3, we need to have that

$$\left(-2\lambda + \frac{\lambda^2\eta_t}{1-\bar{\alpha}_t} + 2\right)\eta_t d \leq 0$$

Due to $d > 0, \eta_t \geq 0$ and $\lambda > 0$, we have that

$$-2\lambda + \frac{\lambda^2\eta_t}{1-\bar{\alpha}_t} + 2 \leq 0 \Leftrightarrow \eta_t \leq \frac{2(\lambda-1)}{\lambda^2}(1-\bar{\alpha}_t).$$

To ensure $\eta_t \geq 0$, we should have $\lambda \geq 1$. From $\max_t \sqrt{1-\bar{\alpha}_t} = 1$, we can conservatively set

$$\eta_t \leq \frac{2(\lambda-1)}{\lambda^2}.$$

As $g(\lambda) = \frac{2(\lambda-1)}{\lambda^2}$ has its maximum in $\lambda \geq 1$ when $g(2) = \frac{1}{2}$, we have $\eta_t \in [0, \frac{1}{2}]$ when $\lambda = 2$. □

### C.3 Proof of Theorem 4.5

In this section, we present materials related to the proof of Theorem 4.5. For the convergence analysis, we adapt the assumptions and result of [70]. First, we introduce the essential definitions, then we provide the technical lemmas and present a proof of the main theorem. Note that these proofs demonstrate the exponential convergence of ULA under the minimal isoperimetric condition (i.e. the Log-Sobolev inequality), without the need for strict and often impractical assumptions such as log-concavity or boundedness of higher derivatives [70].

#### C.3.1 Definitions

**Definition C.1** (Kullback-Leibler (KL) divergence). The Kullback-Leibler (KL) divergence of $p$ with respect to $q$ is defined as

$$\mathrm{D}_{\mathrm{KL}}(p \parallel q) = \int_{\boldsymbol{x}\in\mathcal{X}\subseteq\mathbb{R}^d} p(\boldsymbol{x})\log\frac{p(\boldsymbol{x})}{q(\boldsymbol{x})}\mathrm{d}\boldsymbol{x}.$$

**Definition C.2** (Log-Sobolev Inequality (LSI)). A probability distribution $p$ satisfies the log-Sobolev inequality with a constant $\gamma > 0$ if for all smooth function $g : \mathbb{R}^d \to \mathbb{R}$ with $\mathbb{E}_p[g^2] < \infty$, and

$$\mathbb{E}_p[g^2\log g^2] - \mathbb{E}_p[g^2]\log\mathbb{E}_p[g^2] \leq \frac{2}{\gamma}\mathbb{E}_p[\|\nabla g\|^2].$$

### C.3.2 Technical Lemmas

In this section, we introduce the essential lemmas and corollaries required to prove the main theorem, i.e., Theorem 4.5. Note that we omit the proofs of adapted lemmas and refer to the original paper cited, i.e., Lemma C.3, Lemma C.7, and Lemma C.8 (for the adapted intermediate result).

**Lemma C.3** (Strong convexity and LSI; Corollary 5.11 of [110]). *Let $\mu$ be a probability measure on $\mathbb{R}^d$ of the form $\mathrm{d}\mu = \exp(-h(\boldsymbol{x}))\mathrm{d}\boldsymbol{x}$. If $h$ satisfies $\nabla_{\boldsymbol{x}}^2 h(\boldsymbol{x}) \geq \gamma \mathbf{I}_d$ for some $\gamma > 0$ then $\mu$ satisfies the LSI with a constant $\gamma$.*

**Corollary C.4** (LSI of well-trained diffusion models with non-expansiveness guarantee). *A well-trained diffusion model with L2 norm-driven energy-based reparameterization as in Eq. (11) and Lemma 4.4 satisfies LSI with constant $\frac{2}{1-\bar{\alpha}_t}$.*

*Proof.* For a well-trained diffusion model, we have $p_{\boldsymbol{\theta}}(\boldsymbol{x}_t, t) \propto \exp(-\frac{\lambda}{2}\|\boldsymbol{\epsilon}_{\boldsymbol{\theta}}(\boldsymbol{x}_t, t)\|_2^2)$ from Eq. (11). With the property of well-trained diffusion models stated in Remark 4.2, we have that

$$
\nabla_{\boldsymbol{x}_t}^2 \left( \frac{\lambda}{2}\|\boldsymbol{\epsilon}_{\boldsymbol{\theta}}(\boldsymbol{x}_t, t)\|_2^2 \right)
$$

$$
= \nabla_{\boldsymbol{x}_t} \left( \frac{\lambda}{2}\nabla_{\boldsymbol{x}_t}\|\boldsymbol{\epsilon}_{\boldsymbol{\theta}}(\boldsymbol{x}_t, t)\|_2^2 \right) = \nabla_{\boldsymbol{x}_t} \left( \lambda\boldsymbol{\epsilon}_{\boldsymbol{\theta}}(\boldsymbol{x}_t, t)^\mathsf{T}\nabla_{\boldsymbol{x}_t}\boldsymbol{\epsilon}_{\boldsymbol{\theta}}(\boldsymbol{x}_t, t) \right) = \nabla_{\boldsymbol{x}_t} \left( \frac{\lambda}{\sqrt{1-\bar{\alpha}_t}}\boldsymbol{\epsilon}_{\boldsymbol{\theta}}(\boldsymbol{x}_t, t) \right)
$$

$$
= \frac{\lambda}{\sqrt{1-\bar{\alpha}_t}}\nabla_{\boldsymbol{x}_t}\boldsymbol{\epsilon}_{\boldsymbol{\theta}}(\boldsymbol{x}_t, t) = \frac{\lambda}{\sqrt{1-\bar{\alpha}_t}} \cdot \frac{1}{\sqrt{1-\bar{\alpha}_t}}\mathbf{I}_d = \frac{\lambda}{1-\bar{\alpha}_t}\mathbf{I}_d.
$$

Due to Lemma 4.4 and Lemma C.3, the LSI constant $\gamma$ of well-trained diffusion models with L2 norm-driven energy-based reparameterization that guarantees non-expansiveness w.r.t. L2 norm is given as $\gamma = \frac{2}{1-\bar{\alpha}_t}$ since $\lambda = 2$. $\square$

**Corollary C.5** (Lipschitz smoothness of an energy function of well-trained diffusion models with non-expansiveness guarantee). *A well-trained diffusion model with L2 norm-driven energy-based reparameterization is $\frac{2}{1-\bar{\alpha}_t}$-Lipschitz smooth.*

*Proof.* It is directly implied from Corollary C.4. $\square$

**Assumption C.6** (Bounded dissimilarity). The pairwise chi-squared divergence between two different local distributions is uniformly bounded by $\kappa$, $\sup_{i\neq j\in[K]} \chi^2(p_i \parallel p_j) < \kappa < \infty$.

**Lemma C.7** (LSI constant of a mixture of distributions; Theorem 1 of [111]). *Denote a mixture of distributions $p^\star := \sum_{i=1}^K w_i p_i$ for $w_i \geq 0, \sum_{i=1}^K w_i = 1$, where each $p_i$ satisfies the LSI with $\gamma_i$. If Assumption C.6 holds, then $p^\star$ also satisfies LSI with a constant of*

$$
\gamma^\star = \frac{\min_{i\in[K]} \gamma_i}{3(1+\kappa)(1+\log(1+\kappa))}
$$

**Lemma C.8** (One-step contraction of ULA; Lemma 3 of [70]). *Let $\boldsymbol{x}_t \sim \tilde{p}_t$ be the output iterate one-step ULA. In one step, ULA can sample from a distribution $p_t \equiv p_{\boldsymbol{\theta}}(\cdot, t)$ encoded by a single well-trained diffusion model, satisfying*

$$
\mathrm{D}_{\mathrm{KL}}(\tilde{p}_{t+1} \parallel p_{t+1}) \leq \exp\left( \left( 8\varsigma^2 - \frac{3}{2} \right)\gamma_t\eta_t \right)\mathrm{D}_{\mathrm{KL}}(\tilde{p}_t \parallel p_t) + 6d\varsigma\eta_t, \tag{A3}
$$

*with step size $0 < \eta_t \leq \frac{\varsigma\gamma_t}{L_t^{p+1}}$, where $L_t$ is Lipschitz smoothness constant, $\gamma_t$ is LSI constant, $p \geq 1$ and $0 < \varsigma < \frac{\sqrt{3}}{4}$.*

*Proof.* Consider the continuous interpolation $\tilde{p}_\tau$, where $\tau \in [0, \eta_t]$ with

$$
\tilde{p}_{\tau=0} = \tilde{p}_t, \qquad \tilde{p}_{\tau=\eta_t} = \tilde{p}_{t+1}. \tag{A4}
$$

Denote the LSI constant of a distribution encoded by well-trained diffusion models as $\gamma_t$ and the Lipschitz smoothness constant as $L_t$. For all $\tau \in [0, \eta_t]$, we can directly adapt the intermediate result of Lemma 3 of [70] as

$$
\frac{\mathrm{d}}{\mathrm{d}\tau}\mathrm{D}_{\mathrm{KL}}(\tilde{p}_\tau \parallel p_\tau) \leq -\frac{3\gamma_\tau}{2}\mathrm{D}_{\mathrm{KL}}(\tilde{p}_\tau \parallel p_\tau) + \frac{4\tau^2 L_\tau^4}{\gamma_\tau}\mathrm{D}_{\mathrm{KL}}(\tilde{p}_0 \parallel p_0) + 2d\tau^2 L_\tau^3 + 2d\tau L_\tau^2.
$$

Denote
$$A_\tau := \frac{4\tau^2 L_\tau^4}{\gamma_\tau} \, D_{KL} \left( \tilde{p}_0 \parallel p_0 \right) + 2d\tau^2 L_\tau^3 + 2d\tau L_\tau^2,$$
and introduce the integrating factor as
$$\mu(\tau) = \exp \left( \frac{3}{2} \int_0^\tau \gamma_s ds \right).$$

We wish to integrate over $\tau = 0$ to $\tau = \eta_t$, thus we have $\tau \le \eta_t$. Further assume that for any $\tau \in [0, \eta_t]$ we have
$$\gamma_t \le \gamma_\tau \le L_\tau \le L_t \tag{A5}$$
Then, we can upper bound as
$$A_\tau \le \frac{4\eta_t^2 L_t^4}{\gamma_t} \, D_{KL} \left( \tilde{p}_0 \parallel p_0 \right) + 2d\eta_t^2 L_t^3 + 2d\eta_t L_t^2 := A_t, \tag{A6}$$
as it becomes irrelevant to $\tau$.

Then, we can rewrite the inequality as
$$\frac{d}{d\tau} \left( \mu(\tau) \, D_{KL} \left( \tilde{p}_\tau \parallel p_\tau \right) \right) \le \mu(\tau) A_t.$$

Integrating this inequality from $\tau = 0$ to $\tau = \eta_t$, we have that
$$\mu(\eta_t) \, D_{KL} \left( \tilde{p}_{\eta_t} \parallel p_{\eta_t} \right) - D_{KL} \left( \tilde{p}_0 \parallel p_0 \right)$$
$$\le A_t \int_0^{\eta_t} \mu(\tau) d\tau = A_t \int_0^{\eta_t} \exp \left( \frac{3}{2} \int_0^\tau \gamma_s ds \right) d\tau$$

Rearranging, we have that
$$D_{KL} \left( \tilde{p}_{\eta_t} \parallel p_{\eta_t} \right) \le \exp \left( -\frac{3}{2} \int_0^{\eta_t} \gamma_\tau d\tau \right) D_{KL} \left( \tilde{p}_0 \parallel p_0 \right)$$
$$+ A_t \exp \left( -\frac{3}{2} \int_0^{\eta_t} \gamma_\tau d\tau \right) \int_0^{\eta_t} \exp \left( \frac{3}{2} \int_0^\tau \gamma_s ds \right) d\tau$$
$$\le \exp \left( -\frac{3}{2} \gamma_t \int_0^{\eta_t} d\tau \right) D_{KL} \left( \tilde{p}_0 \parallel p_0 \right)$$
$$+ A_t \exp \left( -\frac{3}{2} \gamma_t \int_0^{\eta_t} d\tau \right) \int_0^{\eta_t} \exp \left( \frac{3}{2} L_t \tau \right) d\tau$$
$$= \exp \left( -\frac{3}{2} \gamma_t \eta_t \right) D_{KL} \left( \tilde{p}_0 \parallel p_0 \right)$$
$$+ A_t \exp \left( -\frac{3}{2} \gamma_t \eta_t \right) \frac{2}{3L_t} \left( \exp \left( \frac{3}{2} L_t \eta_t \right) - 1 \right),$$
where the second inequality is due to Eq. (A5).

Using the inequality that $e^c \le 1 + 2c$ for $0 < c = \frac{3}{2} L_t \eta_t \le 1$ (which holds due to the assumption that $\eta_t \le \frac{\varsigma}{L_t^p} \le \frac{2}{3L_t}$) along with Eq. (A6) we have that
$$D_{KL} \left( \tilde{p}_{\eta_t} \parallel p_{\eta_t} \right)$$
$$\le \exp \left( -\frac{3}{2} \gamma_t \eta_t \right) D_{KL} \left( \tilde{p}_0 \parallel p_0 \right) + A_t \exp \left( -\frac{3}{2} \gamma_t \eta_t \right) \cdot 2\eta_t$$
$$= \exp \left( -\frac{3}{2} \gamma_t \eta_t \right) \left( 1 + \frac{8\eta_t^3 L_t^4}{\gamma_t} \right) D_{KL} \left( \tilde{p}_0 \parallel p_0 \right)$$
$$+ \exp \left( -\frac{3}{2} \gamma_t \eta_t \right) \left( 4d\eta_t^3 L_t^3 + 4d\eta_t^2 L_t^2 \right)$$
$$\le \exp \left( -\frac{3}{2} \gamma_t \eta_t \right) \left( 1 + \frac{8\eta_t^3 L_t^4}{\gamma_t} \right) D_{KL} \left( \tilde{p}_0 \parallel p_0 \right) + \left( 4d\eta_t^3 L_t^3 + 4d\eta_t^2 L_t^2 \right),$$

where the last inequality is due to $\exp\left(-\frac{3}{2}\gamma_t\eta_t\right) \leq 1$.

Using the assumption that $\eta_t \leq \frac{\varsigma\gamma_t}{L_t^{p+1}} \leq \frac{\varsigma}{L_t^p}$, we have

$$1 + \frac{8\eta_t^3 L_t^4}{\gamma_t} \leq 1 + \frac{8\varsigma\eta_t^2 L_t^{4-p}}{\gamma_t} \leq 1 + 8\varsigma^2\gamma_t\eta_t \leq \exp\left(8\varsigma^2\gamma_t\eta_t\right).$$

Thus, the inequality above becomes that

$$\begin{aligned}
\mathrm{D}_{\mathrm{KL}}\left(\tilde{p}_{\eta_t} \parallel p_{\eta_t}\right) &\leq \exp\left(-\frac{3}{2}\gamma_t\eta_t\right)\exp\left(8\varsigma^2\gamma_t\eta_t\right)\mathrm{D}_{\mathrm{KL}}\left(\tilde{p}_0 \parallel p_0\right) + 4d\eta_t^3 L_t^3 + 4d\eta_t^2 L_t^2 \\
&= \exp\left(-\frac{3}{2}\gamma_t\eta_t\right)\exp\left(8\varsigma^2\gamma_t\eta_t\right)\mathrm{D}_{\mathrm{KL}}\left(\tilde{p}_0 \parallel p_0\right) + 4d\eta_t^2 L_t^2(\eta_t L_t + 1) \\
&= \exp\left((8\varsigma^2 - \gamma_t)\eta_t\right)\mathrm{D}_{\mathrm{KL}}\left(\tilde{p}_0 \parallel p_0\right) + 4d\eta_t^2 L_t^2(\eta_t L_t + 1)
\end{aligned}$$

As $\eta_t \leq \frac{\varsigma}{L_t^p} \leq \frac{1}{2L_t}$ for $p \geq 1$, we have $\eta_t L_t \leq \frac{1}{2}$ and $\eta_t L_t^2 \leq \varsigma$

$$\begin{aligned}
\mathrm{D}_{\mathrm{KL}}\left(\tilde{p}_{\eta_t} \parallel p_{\eta_t}\right) &\leq \exp\left(\left(8\varsigma^2 - \frac{3}{2}\right)\gamma_t\eta_t\right)\mathrm{D}_{\mathrm{KL}}\left(\tilde{p}_0 \parallel p_0\right) + 4d\eta_t^2 L_t^2(\eta_t L_t + 1) \\
&\leq \exp\left(\left(8\varsigma^2 - \frac{3}{2}\right)\gamma_t\eta_t\right)\mathrm{D}_{\mathrm{KL}}\left(\tilde{p}_0 \parallel p_0\right) + 6d\varsigma\eta_t.
\end{aligned}$$

Finally, replacing with Eq. (A4), we finally have that

$$\mathrm{D}_{\mathrm{KL}}\left(\tilde{p}_{t+1} \parallel p_{t+1}\right) \leq \exp\left(\left(8\varsigma^2 - \frac{3}{2}\right)\gamma_t\eta_t\right)\mathrm{D}_{\mathrm{KL}}\left(\tilde{p}_t \parallel p_t\right) + 6d\varsigma\eta_t. \tag{A7}$$

$\square$

**Lemma C.9** (Convergence of ULA in KL divergence). *Let $p_t \equiv p_{\boldsymbol{\theta}}(\cdot, t)$ be the probability distribution defined by a single well-trained diffusion model and let $B_\varsigma = \frac{3}{2} - 8\varsigma^2 > 0$. Assume that the iterates $\boldsymbol{x}_t \sim \tilde{p}_t$ are generated by the Unadjusted Langevin Algorithm (ULA) in Eq. (8), and that $\mathrm{D}_{\mathrm{KL}}(\tilde{p}_0 \parallel p_0) < \infty$. Then, for all $t \geq 0$, we have*

$$\mathrm{D}_{\mathrm{KL}}(\tilde{p}_t \parallel p_t) \leq \exp\left(-B_\varsigma t\gamma_t\eta_0\right)\mathrm{D}_{\mathrm{KL}}(\tilde{p}_0 \parallel p_0) + \frac{9d\varsigma\eta_t}{B_\varsigma\gamma_t\eta_0}. \tag{A8}$$

*Hence, for any $\delta \geq \frac{18d\varsigma}{B_\varsigma\gamma_t}$, it suffices to run ULA for*

$$t \geq \frac{1}{B_\varsigma\gamma_t\eta_0}\log\left(\frac{2\,\mathrm{D}_{\mathrm{KL}}(\tilde{p}_0 \parallel p_0)}{\delta}\right)$$

*steps with step size*

$$\eta_t \leq \min\left\{\frac{B_\varsigma\gamma_t\eta_0\delta}{18d\varsigma}, \frac{\varsigma\gamma_t}{L_t^{p+1}}\right\},$$

*for $p \geq 1$ and LSI constant $\gamma_t$, in order to guarantee $\mathrm{D}_{\mathrm{KL}}(\tilde{p}_t \parallel p_t) \leq \delta$.*

*Proof.* From Lemma C.8, recursively applying Eq. (A7) gives

$$\mathrm{D}_{\mathrm{KL}}(\tilde{p}_t \parallel p_t) \leq \exp\left(-B_\varsigma \sum_{s=0}^{t-1}\gamma_s\eta_s\right)\mathrm{D}_{\mathrm{KL}}(\tilde{p}_0 \parallel p_0) + 6d\varsigma\sum_{r=0}^{t-1}\eta_r\exp\left(-B_\varsigma\sum_{s=r+1}^{t-1}\gamma_s\eta_s\right).$$

Since we have $\eta_t \geq \cdots \geq \eta_0$ and $\gamma_0 \geq \cdots \geq \gamma_t$, we can bound that

$$\sum_{s=0}^{t-1}\gamma_s\eta_s \geq t\gamma_t\eta_0, \quad \sum_{s=r+1}^{t-1}\gamma_s\eta_s \geq (t-r-1)\gamma_t\eta_0.$$

Because $B_\varsigma > 0$, we get:

$$D_{KL}(\tilde{p}_t \parallel p_t) \leq \exp(-B_\varsigma t\gamma_t\eta_0) D_{KL}(\tilde{p}_0 \parallel p_0) + 6d\varsigma \sum_{r=1}^{t} \eta_t \exp(-B_\varsigma r\gamma_t\eta_0).$$

The remaining sum is for a geometric series, thus

$$\sum_{r=1}^{t} \exp(-B_\varsigma r\gamma_t\eta_0) = \exp(-B_\varsigma \gamma_t\eta_0) \cdot \frac{1 - \exp(-B_\varsigma \gamma_t\eta_0 t)}{1 - \exp(-B_\varsigma \gamma_t\eta_0)}$$

$$\leq \frac{1}{1 - \exp(-B_\varsigma \gamma_t\eta_0)} \leq \frac{3}{2B_\varsigma \gamma_t\eta_0},$$

where the last inequality uses

$$\frac{2c}{3} \leq 1 - e^{-c}, \quad 0 < c = B_\varsigma \gamma_t\eta_0 \leq \varsigma < \frac{\sqrt{3}}{4},$$

from Lemma C.8.

Thus,

$$D_{KL}(\tilde{p}_t \parallel p_t) \leq \exp(-B_\varsigma t\gamma_t\eta_0) D_{KL}(\tilde{p}_0 \parallel p_0) + \frac{9d\varsigma\eta_t}{B_\varsigma \gamma_t\eta_0}.$$

To ensure $D_{KL}(\tilde{p}_t \parallel p_t) \leq \delta$, it suffices to assume:

$$\frac{9d\varsigma\eta_t}{B_\varsigma \gamma_t\eta_0} \leq \tfrac{\delta}{2}, \quad \exp(-B_\varsigma t\gamma_t\eta_0) D_{KL}(\tilde{p}_0 \parallel p_0) \leq \tfrac{\delta}{2},$$

which hold when

$$\eta_t \leq \frac{B_\varsigma \gamma_t\eta_0\delta}{18d\varsigma} \quad \text{and} \quad t \geq \frac{1}{B_\varsigma \gamma_t\eta_0} \log\left(\frac{2 D_{KL}(\tilde{p}_0 \parallel p_0)}{\delta}\right).$$

$\square$

### C.3.3 Proof of Theorem 4.5

Denote $p_{ti} \equiv p_{\boldsymbol{\theta}_i}(\cdot, t)$ as a distribution encoded by a locally-trained diffusion model of client $i$. For the mixture of distribution $p_t^\star = \sum_{i=1}^{K} w_i p_{ti}$, it is trivial that the energy function of $p_t^\star$ is $L_t^\star = \frac{2}{1-\bar{\alpha}_t}$-Lipschitz smooth since each local distribution is Lipschitz smooth due to Corollary C.5.

From the result of Lemma C.7, the LSI constant of the mixture is $\gamma^\star = \frac{\min_{i\in[K]} \gamma_i}{3(1+\kappa)(1+\log(1+\kappa))}$. As each local distribution has LSI constant $\gamma_{ti} = \frac{2}{1-\bar{\alpha}_t}$, we can further refine as

$$\gamma_t^\star = \frac{2}{3(1 - \bar{\alpha}_t)(1 + \kappa)(1 + \log(1 + \kappa))}.$$

Denote $0 < \tilde{\kappa} = \frac{2}{3(1+\kappa)(1+\log(1+\kappa))} < \frac{2}{3}$, we set the LSI constant as $\gamma_t^\star = \frac{\tilde{\kappa}}{1-\bar{\alpha}_t}$.

From the result of Lemma C.9, we finally have that

$$D_{KL}(\tilde{p}_t^\star \parallel p_t^\star) \leq \exp\left(-B_\varsigma t\gamma_t^\star\eta_0\right) D_{KL}(\tilde{p}_0^\star \parallel p_0^\star) + \frac{9d\varsigma\eta_t}{B_\varsigma \gamma_t^\star\eta_0}$$

$$= \exp\left(-\frac{B_\varsigma \tilde{\kappa}\eta_0 t}{1 - \bar{\alpha}_t}\right) D_{KL}(\tilde{p}_0^\star \parallel p_0^\star) + \frac{9d\varsigma\eta_t(1 - \bar{\alpha}_t)}{B_\varsigma \tilde{\kappa}\eta_0}$$

with step size

$$\eta_t \leq \min\left\{\frac{B_\varsigma \gamma_t^\star\eta_0\delta}{18d\varsigma}, \frac{\varsigma\gamma_t^\star}{(L_t^\star)^{p+1}}\right\} = \min\left\{\frac{B_\varsigma \tilde{\kappa}\eta_0\delta}{18d\varsigma(1 - \bar{\alpha}_t)}, \frac{\varsigma\tilde{\kappa}(1 - \bar{\alpha}_t)^p}{2^p}\right\}.$$

In practice, however, we cannot directly quantify $\bar{\kappa}$. Thus, we instead manually adjust a constant $\rho := \varsigma\bar{\kappa} < \frac{\sqrt{3}}{6}$. Further denote $\upsilon := B_\varsigma \tilde{\kappa}\eta_0$.

Finally, we have that

$$D_{\mathrm{KL}}(\tilde{p}_t^\star \| p_t^\star) \leq \exp\left(-\frac{\upsilon t}{1-\bar{\alpha}_t}\right) D_{\mathrm{KL}}(\tilde{p}_0^\star \| p_0^\star) + \frac{9d_\varsigma \eta_t (1-\bar{\alpha}_t)}{\upsilon},$$

with step size

$$\eta_t \leq \min\left\{\frac{\upsilon\delta}{18d_\varsigma(1-\bar{\alpha}_t)}, \frac{\rho(1-\bar{\alpha}_t)^p}{2^p}\right\},$$

for any $\delta \geq \frac{18d_\varsigma \eta_0(1-\bar{\alpha}_t)}{\upsilon}$ in $t \geq \frac{1-\bar{\alpha}_t}{\upsilon} \log\left(\frac{2\,D_{\mathrm{KL}}(\tilde{p}_0 \| p_0)}{\delta}\right)$ steps. Finally, replacing $0 \to T$ and $t \to T - t$ for the compatibility with ULA reaches the theorem statement.

# D  Experimental Details

**Specification.**  We conduct all experiments in a single server with Intel® Xeon® Gold 6226R CPU (@ 2.90GHz) and a single NVIDIA® Ampere® A100 GPU (w/ 40GB VRAM). For the implementation of diffusion models, we resort to `diffusers` [112] library using `PyTorch` [113].

**Simulation of Statistical Heterogeneity.**  For the faithful evaluation of practical FL setting, we simulate non-IID data split to $K = 10$ clients for all benchmark datasets.

For MNIST dataset, we use Dirichlet distribution with concentration parameter $\alpha = 0.1$, following the setting of [79].

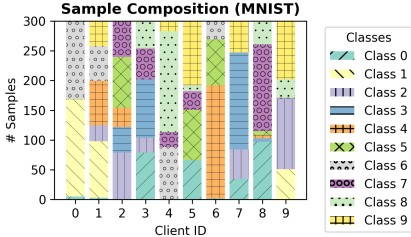

Figure A1: Non-IID local distributions of MNIST dataset

For CIFAR-10 dataset, we follow the setting of [21] using log-normal distribution with location=0 and scale=2.

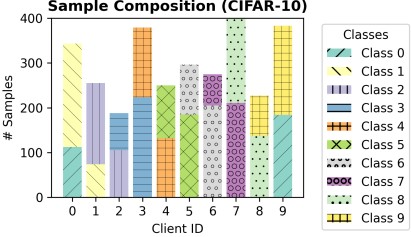

Figure A2: Non-IID local distributions of CIFAR-10 dataset

For CelebA dataset, which has 40 different attributes, we first construct classes by combining gender (male/female), smiling (0/1), and eyeglasses (0/1) attributes, i.e., 8 classes as a result. We randomly distribute samples to clients so that they have only three distinct classes.

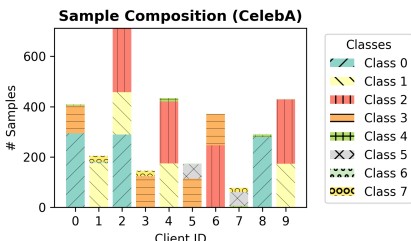

Figure A3: Non-IID local distributions of CelebA dataset

**Model and Training Hyperparameters.** We summarize detailed configurations of models used for experiments in Table A1.

Table A1: Model and Training Configurations.

|  | MNIST | CIFAR-10 | CelebA |
|---|---|---|---|
| *Model Configuration* | | | |
| Spatial dimension | | $32 \times 32$ | |
| Attention resolution | | $8 \times 8$ | |
| Base channels | | 128 | |
| Channel multipliers | 1, 1, 1, 1 | 1, 2, 2, 2 | |
| Model size | 44.77MB | 136.38MB | 136.38MB |
| Base architecture | | DDPM [9] | |
| Scheduling scheme | | linear scheduling [9] | |
| *Training Configuration* | | | |
| Optimizer | | Adam [114] | |
| Learning rate | | $2 \times 10^{-4}$ | |

