# OpenReview forum: "Diffusion Federated Dataset"
_NeurIPS.cc/2025/Conference — NeurIPS 2025 poster_

### Official Review · Reviewer_1TqA · 2025-06-30

**Clarity:** 3
**Significance:** 4
**Originality:** 4
**Rating:** 5
**Confidence:** 4

**Summary:**

This paper proposes COODI, a novel framework for federated synthetic data generation based on cooperative inference of diffusion models. Rather than aggregating model parameters during training, COODI reframes federated generation as a sampling problem over a mixture of locally trained diffusion models. Each client independently trains a conditional diffusion model on its private data, and the server later coordinates the sampling process by requesting noise predictions from clients and aggregating their outputs into a global mixture score. The sampling follows an Unadjusted Langevin Algorithm (ULA) guided by these aggregated scores.

In addition, the authors provide a theoretical analysis connecting diffusion models to energy-based models (EBMs), derive convergence guarantees under statistical heterogeneity, and demonstrate strong empirical results on multiple image datasets. COODI achieves high data fidelity, utility, and privacy preservation, offering a communication-efficient solution suitable for cross-silo federated settings.

**Questions:**

Please refer to Weakness for details.

**Ethical Concerns:**

["NO or VERY MINOR ethics concerns only"]

**Final Justification:**

After reading the authors’ rebuttal and  clarifications, I think most of my main concerns have been addressed—especially around how CooDI might be improved and what the final outputs are meant to be. The authors were transparent about the current limitations and shared some good ideas for future extensions. That said, I feel that the need for full client synchronization is still a real drawback and limits the method’s practical use right now. Overall, I see this as a valuable step forward for federated dataset distillation in cross-silo setups.

**Limitations:**

yes

**Quality:**

4

**Strengths And Weaknesses:**

Strenths:

1. The paper introduces a new perspective on federated synthetic data generation by reframing the problem from a model-centric training paradigm to a data-centric cooperative inference strategy. Rather than performing expensive and communication-intensive federated training of generative models, COODI proposes a collaborative sampling framework wherein clients locally train their own diffusion models and participate only in the inference stage. The central server aggregates their outputs to perform mixture-based Langevin sampling. This strategy enables a more modular and communication-efficient system design, particularly well-suited for cross-silo settings where each participant retains full control of its local model and data.

2. I believe that a key strength of the paper lies in its rigorous theoretical foundation. The authors establish a clear connection between diffusion models and energy-based models (EBMs), leveraging this relationship to implement Unadjusted Langevin Algorithm (ULA) sampling without requiring access to model parameters or normalized probability densities. This interpretation allows the system to operate within a well-understood probabilistic framework. Furthermore, the authors derive non-asymptotic convergence guarantees under statistical heterogeneity, which is particularly important in the federated setting where client distributions may differ significantly. These results lend both credibility and depth to the proposed method.

3.  The experimental section is extensive and well-executed. The authors evaluate COODI across a wide range of datasets, including CIFAR-10, CelebA, MNIST, and etc. The results show clear improvements in generation quality and in the utility of synthetic data when used for downstream classification tasks.

Weaknesses
1. The current framework appears to require all clients to synchronously participate in each sampling step by providing noise predictions for the server to compute the global mixture score. This design might limit applicability in more dynamic or large-scale cross-device federated learning scenarios, where clients are often intermittently available. While this constraint seems acceptable in cross-silo settings, could the authors clarify whether asynchronous or partially-participating variants of COODI are feasible, or whether such extensions have been explored?

2. The paper emphasizes the generation of synthetic data through collaborative sampling, but it is somewhat unclear what the final deliverable is. Does COODI aim to produce a reusable global generative model (e.g., via score distillation), or are the synthetic samples themselves intended to be directly used for downstream tasks? An explicit clarification of the intended output and deployment scenario would help contextualize the practical usage of the method.

---

> ### Author Rebuttal · Authors · 2025-07-27
>
> Dear **reviewer 1TqA**,
> We deeply appreciate the reviewer's accurate summary of our work as well as the recognition of the significance and strengths of `CooDI`.
> In the following sections, we address the points identified as weaknesses.
>
> ---
> ### 1) On the feasibility of improved `CooDI`
> We appreciate the reviewer's feedback and agree that the suggested improvements are valid.
> - As stated in the contribution (lines 55-56) and limitation (line 280) sections, `CooDI` is designed for a cross-silo FL setting, due to its synchronization requirement from all clients.
>   - However, we believe that `CooDI` can be improved in terms of the scalability, as we have conceptualized in lines 280-282.
> - To the best of our knowledge, no such extensions (e.g., asynchronous variant or ULA with partially observed _scores_) are currently available. Instead, a paper proposed ULA with partially observed _parameters_ under FL setting [1].
>   - In line with this, specifically for the partial participation setting, `CooDI` can be modified to construct an unbiased estimator of the full mixture score based on the observed local scores from partially available clients at each round, similarly to [1].
> - We are open and eager to extend our proposed framework in this direction as future work.
>
> ### 2) Final deliverables from `CooDI`
> We thank the reviewer for highlighting the ambiguous part of our manuscript. We will revise our manuscript accordingly to ensure clear delivery.
> - Originally, we intended that the _synthetic dataset themselves_ are worth to obtain, as it can be explicitly used to train a server-side global model for downstream task.
>   - That's why we designed an experiment to evaluate the _utility_ perspective of the synthetic dataset. (lines 258-259)
> - We also believe that a synthetic dataset on a credible central server would benefit the following:
>   - i) incoming clients in the FL system;
>   - ii) clients with data deficiency or imbalance;
>   - iii) the server to manage data quality or protocols (e.g., data standardization) by using the synthetic dataset as a proxy for the collection of local datasets.
> - Alternatively, similar to the score distillation approach mentioned by the reviewer, we are also open to improving `CooDI` to produce a reusable, server-side generative model, based on securing amortized sampler at the server (lines 285-287; e.g., [2]).
> ---
> Again, we sincerely thank the **reviewer 1TqA** for insightful comments.
>
> [1] On Convergence of Federated Averaging Langevin Dynamics (Deng et al., 2021)
> [2] Learning Energy-Based Models by Cooperative Diffusion Recovery Likelihood (Zhu et al., 2023)

---

> > ### Comment · Reviewer_1TqA · 2025-08-04
> > **Thanks for the rebuttal**
> >
> > I have reviewed the rebuttal and found that all my questions have been thoroughly addressed.

---

> > > ### Author Response · Authors · 2025-08-04
> > >
> > > We appreciate the positive evaluation and comments from the reviewer. Thank you for all your efforts and service!

---

> > > > ### Comment · Reviewer_1TqA · 2025-08-05
> > > >
> > > > Please consider including the discussion in the final version. Thanks.

---

### Official Review · Reviewer_u5Vy · 2025-06-30

**Clarity:** 3
**Significance:** 3
**Originality:** 4
**Rating:** 4
**Confidence:** 5

**Summary:**

This paper presents CooDI, a diffusion model-based federated learning method. The method requires all clients to first train local diffusion models independently. Subsequently, the server initializes multiple noise vectors with corresponding prompts. During the denoising process, the server distributes current-step samples to all clients and collects their denoising results for these samples. The server then aggregates outputs from all clients and completes the current-step denoising based on the aggregated results. After T iterations, the server obtains a complete synthetic dataset. Experiments on MNIST, CIFAR-10, and CelebA datasets validate the proposed method's performance.

**Questions:**

See weaknesses.

**Ethical Concerns:**

["NO or VERY MINOR ethics concerns only"]

**Final Justification:**

Although I still believe that the proposed method has certain practical limitations, the paper demonstrates sufficient novelty and theoretical contribution. Therefore, I have decided to raise my score.

**Limitations:**

The primary limitations of the proposed method lie in its computational and communication overhead.

**Paper Formatting Concerns:**

N/A.

**Quality:**

2

**Strengths And Weaknesses:**

Strengths:
1. The introduction of Energy-based Diffusion Models (EBMs) perspective into federated learning demonstrates notable innovation.
2. The paper provides formal proofs for both privacy preservation and convergence in diffusion model-based federated learning, establishing theoretical soundness.

Weaknesses:
1. A fundamental contradiction exists in the methodology: While Line 30 explicitly states that diffusion models are generally unsuitable for federated learning due to their massive parameter size (which increases computational and communication overhead), the proposed method not only employs diffusion models but further exacerbates these issues by requiring client-side fine-tuning and multiple iterations. This represents a significant methodological paradox that undermines the approach's practicality.
2. The experimental comparison is limited to only MD-GAN and PRISM methods, failing to include other state-of-the-art diffusion model-based federated learning approaches. This limited baseline comparison substantially weakens the experimental validity.
3. Several critical experimental details remain unspecified: The exact architecture of client-side diffusion models is unclear. The claim of using "DDPM" is problematic as it refers to a sampling scheduler rather than a specific model architecture.
4. The evaluation solely focuses on synthetic dataset quality, which deviates from standard federated learning benchmarks that typically assess downstream tasks (e.g., image classification). Moreover, the inherent randomness in diffusion generation processes suggests potential high variance in experimental results, yet no variance analysis is provided.

---

> ### Author Rebuttal · Authors · 2025-07-28
>
> Dear  **reviewer u5Vy**,
> We appreciate your thoughtful reviews. We have addressed each of your concerns in detail below.
>
> ----------
> ### 1) Practicality of  `CooDI`
> We understand the concern, though we believe there may be a misinterpretation. Below, we clarify the relevant aspects:
> - i) `CooDI` does **not require the exchange of diffusion model parameters.**
>   - Training a diffusion model through repetitive parameter exchange, as is done in traditional FL, is indeed unsuitable due to the computational and communication overhead. (line 30)
>   - Rather, `CooDI` requires **exchange of synthetic data, not parameters of diffusion models.** (line 6 of Algorithm 1; line 196; lines 179-180; Figure 1-B & C)
>   - That being said, the end goal of `CooDI` is not to obtain a diffusion model. Instead, `CooDI` aims to obtain a global synthetic dataset.
>   - The synthetic data is **far lighter in size than the parameters, as provided in Table 3**, and emphasized in the **Efficiency** subsection. (line 271)
>   - That's why we framed `CooDI` as a _data-centric_ approach in Abstract (line 1) and Introduction (line 24), since `CooDI` can detour repetitive and burdensome parameter exchanges. (line 81; lines 179-180)
> - ii) `CooDI` does **not require client-side fine-tuning for multiple iterations.**
>   - We designed the local training to be required _only once_, right before the communication begins. (lines 184-189; line 3 of Algorithm 1)
>   - `CooDI` **only requires local inference** w.r.t. the collaborative synthetic dataset; no local fine-tuning is required. (lines 195-200; lines 6-7 in Algorithm 1; Figure 1-C)
> - Thus, `CooDI` **does not exacerbate communication/computation issues**, but **rather improves efficiency** compared to parameter exchange-based FL methods for a diffusion model because it **does not necessitate repetitive local fine-tuning nor multiple parameter exchanges**.
>
> ### 2) Comparison with SotA federated diffusion approach
> We thank the reviewer for pointing out concern on experimental baselines.
> - Originally, we regard `CooDI` as a _federated synthetic data generation_ framework, not a federated diffusion modeling method. (lines 34-38; lines 45-50)
>   - That's why we focused on previous works on **synthetic data generation methods under federated settings**, including `MD-GAN` and `PRISM`.
> - However, we respectfully agree with the reviewer's comment and acknowledge that the federated diffusion model can also be considered an additional baseline, since it can also generate synthetic data at the end.
>   - Thus, we provide additional comparison results on MNIST dataset with state-of-the-art federated diffusion modeling method, i.e., `FedDiffuse` [1], with shards partitioning setting for non-IID and $K=10$ clients in the table below.
> |  | FID | LogReg (Acc.) | MLP (Acc.) | CNN (Acc.) |
> |:--|:--:|:--:|:--:|:--:|
> | `FedDiffuse` | 58.4 | 81.2 | 81.1 | 82.9 |
> | `CooDI` | **4.8730** | **92.8** | **91.2** | **93.9** |
>   -  Note that we exploited `full` strategy with best reported hyperparameter in [1]. `CooDI` consistently outperforms in both synthetic data quality and utility, even under practical non-IID setting.
>
> ### 3) Experimental Details
> - We mentioned (line 259) and readily provided the exact architecture of client-side diffusion models (**Table A1** in the technical appendix), with the source code attached in the supplemental material.
> - To clarify again, we adopted the U-Net architecture [2,3], which is typically used in diffusion modeling and is described in the appendix, using `diffusers` package.
> - Due to the space limit, we also separately spared _Experimental Details_ section in the technical appendix that contains several other experimental details.
>
> ### 4) Evaluation of Synthetic Dataset Utility
> - We've already provided **image classification results on a synthetic dataset in Table 2** as one of our main results, in order to evaluate the utility of synthetic dataset for a downstream task.
> - We appreciate the reviewer’s concern on variance due to the stochastic nature of diffusion models. While we couldn't report variance across multiple random seeds due to resource constraints during the submission, we note that the selected evaluation metrics (FID, Precision & Recall, Density & Coverage) are designed to capture quality and distributional differences over multiple generated images, and are commonly used to benchmark generative model robustness.
> - In our experiments, we observed consistent performance and reasonable stability. Still, we acknowledge that a full variance analysis (e.g., over multiple seeds) would strengthen the evaluation, and we plan to include such results in camera ready version if accepted.
>
> ----------
> Again, we are grateful for the time and effort that **reviewer u5Vy** dedicated to this review.
>
> [1] Training diffusion models with federated learning (de Goede, Cox and Decouchant, 2024)
> [2] U-Net: Convolutional networks for biomedical image segmentation (Ronneberger, Fischer and Brox, 2015)
> [3] Denoising Diffusion Probabilistic Models (Ho, Jain and Abbeel, 2020)

---

> > ### Comment · Reviewer_u5Vy · 2025-08-05
> >
> > I acknowledge the innovation and soundness of the proposed method. However, my primary concern lies in its practical feasibility.
> >
> > While I understand that diffusion model parameters need not be exchanged (only intermediate samples are shared), the client training and inference with diffusion models remain computationally demanding. More critically, each timestep of the denoising process requires aggregation of the predicted noises, leading to frequent client-server communication. This could incur substantial communication overhead, which the authors should analyze more thoroughly.
> >
> > The visual quality of this paper could be significantly enhanced by testing on higher-resolution datasets, which would better demonstrate its real-world applicability and improve the paper’s presentation. Given the remarkable capability of diffusion models in handling high-fidelity, complex data, such simple datasets fail to provide meaningful insights into the model's performance on realistic, challenging cases.

---

> > > ### Author Response · Authors · 2025-08-07
> > >
> > > Dear **Reviewer u5Vy**,
> > >
> > > Thank you again for your valuable feedback. we’ve provided a clarification for your concerns on practicality of our method in our earlier reply.
> > > Since the discussion period ends soon, we would be very grateful if you could review our response and let us know if it resolves your concern or if there’s anything else we can address.
> > >
> > > We truly appreciate your time and engagement.
> > >
> > > Sincerely,
> > > Authors of Paper 25315

---

> ### Author Response · Authors · 2025-08-05
>
> Thank you for your insightful and constructive feedback. We appreciate the opportunity to address your comments regarding the practical feasibility of our proposed method, `CooDI`. Below, we clarify how our target setting and experimental results demonstrate its practicality.
>
> ### `CooDI` is practical as it targets for **cross-silo FL** setting
> - As outlined in our manuscript, `CooDI` is designed for the **cross-silo FL setting** (line 56; lines 278-281), which assumes _a moderate number of reliable, known, and addressable_ clients (e.g., medical institutions, banks, or retail companies) with **sufficient computational resources**, contrary to the resource-constrained clients typical in cross-device FL settings [1].
> - While we acknowledge that diffusion model training can be computationally intensive in cross-device scenarios, this concern is **significantly mitigated in the cross-silo setting** due to the clients’ robust computational capabilities.
> - Additionally, our **analysis of communication and computation overhead (Table 3)** further supports `CooDI`’s efficiency:
>   - i) Communication efficiency: `CooDI` requires **fewer communication bits** compared to traditional parameter exchange-based FL ($N \times d < p$; line 271).
>   - ii) Computation efficiency: Local computation costs during collaboration are **reduced on average**, making `CooDI` well-suited even with the frequent denoising process.
>   - Please further note that `CooDI` also has the potential to improve both perspectives, as discussed in lines 282-285.
> - With these reasons, we humbly believe that `CooDI` proves its practicality, specifically for cross-silo FL setting.
>   - Please also note that this practical aspect is further validated with additional results we describe below.
>
> ### We've already provided additional demonstration of `CooDI` on **real-world, high-resolution datasets** in healthcare
> - To demonstrate the applicability of `CooDI`, **we've already provided results on real-world high-resolution datasets**, considering tasks including _cell segmentation, multi-domain generalization, and MRI classification_ (please see Appendix E, Figures A2–A4).
> - Due to space constraints, these results are included in the supplemental material (lines 273–275).
> - We kindly invite you to review them, as they highlight `CooDI`’s practical utility in a medical consortium setting (i.e., a typical example of cross-silo FL) for the following datasets, **all of which are collected from real-world clinical scene**:
>   - i) **Kvasir** dataset [2]: 96×96 resolution, hospital-collected images, manually annotated by expert gastroenterologists.
>   - ii) **SMDG** dataset [3]: 64×64 resolution, comprising 19 public fundus glaucoma image datasets.
>   - iii) **ISIC-2019** dataset [4] (also a part of the **FLamby** benchmark [5]): 96×96 resolution, featuring skin lesion images from real medical scenarios.
> - These results supports `CooDI`’s **feasibility for real-world applications in cross-silo FL** settings, particularly in healthcare.
>
> We sincerely thank you for your thoughtful feedback, which has allowed us to clarify these points.
> We hope this response addresses your concerns, and we welcome further discussion.
>
>
> ### References
> [1] Advances and Open Problems in Federated Learning (Kairouz et al., 2019)
> [2] Kvasir-seg: A segmented polyp dataset (Jha et al., 2020)
> [3] Smdg, a standardized fundus glaucoma dataset (Kiefer, 2023)
> [4] Skin Lesion Analysis toward Melanoma Detection: A Challenge at the International Symposium on Biomedical Imaging (ISBI) 2016, hosted by the International Skin Imaging Collaboration (ISIC) (Gutman et al., 2016)
> [5] Flamby: Datasets and benchmarks for cross-silo federated learning in realistic healthcare settings (du Terrail et al., 2022)

---

### Official Review · Reviewer_oT3q · 2025-07-02

**Clarity:** 2
**Significance:** 2
**Originality:** 3
**Rating:** 4
**Confidence:** 3

**Summary:**

Synthetic data generation is an interesting task for federated learning (FL). The typical FL approach is for many data holders to compute gradients w.r.t to some model and send them to a centralized server that will aggregate them and send back model updates. As generative models such as LLMs and diffusion models become larger, the computation required to compute local gradients and the communication required to send model parameter updates back and forth become intractable. This work develops the method CooDI which is a FL procedure for sampling data from a mixture of local diffusion data without communicating model parameters or gradients. CooDI consists of two initial steps: each party trains a local diffusion model and the server generates an initial synthetic dataset of random noise. Then these steps are repeated: the server sends the synthetic data to the data holders, the data holders run inference with the local diffusion models, the data holders send the diffusion steps to the server, the server converts the diffusion steps to a score (as in grad logp), then the server weights and combines all the local scores for each record, then the server takes a ULA step with each synthetic record according to the aggregated score. The authors prove several results about this CooDI procedure including a required range for the ULA step size and a convergence result. The authors also evaluate CooDI experimentally and show that it performs well compared to several benchmarks.

**Questions:**

Seeing as the first step of CooDI is for each party to locally train a diffusion model, a simple baseline would be for each party to sample some records from their local model independently and share them with the other parties. This would lead to a synthetic dataset from an equally weighted mixture of each diffusion model. How is the model implied by CooDI different from this simple mixture? How does CooDI compare to the naive mixture experimentally?

What happens if some of the local diffusion models are not “well trained” would CooDI handle this issue gracefully?

DP diffusion model training is not trivial, how much does performance degrade when each client trains a DP diffusion model?

The table in the supplementary material doesn’t seem to include the baseline methods, were the hyperparameters of the baselines tuned?

**Ethical Concerns:**

["NO or VERY MINOR ethics concerns only"]

**Final Justification:**

The authors helped clarify for me the motivation for COODI and I think that if the writing is improved in the manuscript it would merit a score of 4

**Limitations:**

CooDI alleviated the communication burden but it still assumes each party has the ability to train a local diffusion model which is still computationally demanding. This is mentioned in the paper but I believe it should be highlighted more. Also, the authors claim that CooDI complies with HIPPA, I believe this is a legal claim outside the scope of a ML conference and should not be included.

**Paper Formatting Concerns:**

The margin on this paper appears to be wider than the other papers I am reviewing.

**Quality:**

3

**Strengths And Weaknesses:**

Strengths:

The authors seem to have advanced the energy-based parameterization of diffusion models by showing that no modification to the training procedure is required if the model is “well-trained”. This is a bit outside my area of expertise but seems to be a meaningful result.

The empirical results show that CooDI meaningfully improves on the baseline methods across most metrics and several datasets.

Weaknesses:

The manuscript demands a lot of readers. Readers would benefit from some degree of familiarity with diffusion models, energy-based models, the recently discovered connection between energy-based models and diffusion models, federated learning, and a bit of differential privacy. I believe more could be done to make the manuscript more clear to a general ML audience (see the questions section for particular things that seem unclear to me).

The approximation for the score of a diffusion model assumes that the model is “well trained”. It is not clear from the paper if this is a reasonable assumption and no experiments are provided showing that diffusion models typically meet this standard.

---

> ### Author Rebuttal · Authors · 2025-07-29
>
> Dear  **reviewer oT3q**,
> We are grateful for the reviewer's insightful comments and efforts for fully ruminating our work. We've also tried our best to address raised concerns below:
>
> ----------
> ### 1) Clarification on the 'well-trained' assumption & Failure case
> - We intended the term 'well-trained' to be close to 'pre-trained'. That being said, as long as each local diffusion model is (literally) _well-trained_ on the local dataset with sufficiently minimized loss, we are ready to operate `CooDI`.
>   - Therefore, we humbly believe that it is not only reasonable but also practical assumption, and several previous works on the composition of diffusion models accredit this [1-3]; they used pre-(or well-)trained diffusion models for combined sampling of multiple concepts.
> - If it's not met, `CooDI` may fail to generate high-fidelity samples due to low-quality signals. That's why we restrict our setting to a cross-silo setting (line 56), in which moderate number of relatively reliable clients collaborate [4].
>   - To encourage explicit compliance with the 'well-trained' condition in practice, we propose adopting _zero-knowledge proofs_ (line 280), to check and ensure each client faithfully completes the training of local diffusion models.
> - Based on the comment, we will further clarify these points in the revised manuscript.
>
> ### 2) Comparison with the naive mixture
> - The proposed alternative (i.e., sampling from each local distribution independently and collecting the results) corresponds to an _explicit_ sampling scheme from a mixture distribution: i.e., $p^\star (\boldsymbol{x})=\sum\nolimits_{i=1}^K w_i p_i (\boldsymbol{x})$.
>   - This scheme is more appropriate _when if local distributions $p_i(\boldsymbol{x})$ are entirely disjoint_.
>   - Note that this idea is partially aligned with `FedSSyn` [5] - but this method distributes the aggregated samples again to augment local dataset, and then conduct FedAvg.
> - However, in typical FL setup, it is unrealistic to assume such extreme separation among local distributions. It is because FL typically assumes local distributions may be heterogeneous, but with the _bounded_ dissimilarity, rather than arbitrary. (cf., Assumption 7 of [6] Assumption 3 of [7], Assumption 1 of [8])
>   - That is why our theoretical analysis (i.e.,  Lemma C.6; Theorem 1 of [9]) is also based on the assumption that the pairwise chi-squared divergences between local distributions are bounded. This assumption aligns with and captures the "bounded dissimilarity" assumption in probabilistic language.
>   - Accordingly, we didn't include the locally-sample-then-aggregate strategy into baseline, as it doesn't go well with federated settings.
> - That being said, under the realistic assumption for FL, there often exists $\boldsymbol{x}$ for which multiple local distributions assign non-negligible probability, which necessitates advanced sampling strategy.
>   - ULA is a principled and effective tactic, as it explicitly mixes local score functions as in Eq. (8), with inherent stochasticity for an exploration from mode to mode via the noise term in Eq. (7).
>   - However, the explicit mixture sampling might dominate certain $\boldsymbol{x}$ with high joint probability, it often results in _low diversity_ and fails to adapt dynamically to the relative importance of each local distribution.
> - In summary, `CooDI` implicitly samples from a mixture of local distributions (with bounded dissimilarity) via ULA. This process naturally adapts the mixture weights during sampling, guided by the energy-based parameterization proposed in Definition 3.1.
> - We will also update discussion on this in the revised manuscript. We thank the reviewer for pointing out this aspect.
>
> ### 3) DP diffusion model training
> - We agree that the DP training is non-trivial and directly impacts the quality of the synthetic dataset per se.
> - While subsection 4.4 (lines 232–246) demonstrates that achieving DP is theoretically straightforward, the resulting dataset quality under $(\epsilon=10,\delta=10^{-5})$-DP --- though expected --- seems unsatisfactory, compared to non-DP case (cf., Figure A1 on MNIST dataset).
>   - Empirically, we observed that the dynamics of collaborative ULA are affected unfavorably when local diffusion model training is constrained by DP-SGD, even if the quality of locally generated samples with DP is acceptable.
> - Thus, we plan to tone down or even move the privacy part back and further investigate the way of enhancing the compatibility of `CooDI` with DP guarantee. We think it is practically important direction and we are eager to explore this as part of our future work.
>   - Also, we hope this will give us more space to revise the manuscript so that it is clearer and more accessible to general audiences in ML fields.
> - We thank the reviewer for giving us the opportunity to improve presentation better for readers.
>
> ### 4) Others
> - We wholeheartedly thank the reviewer for granular feedback on many other details in our draft.
> - We tuned all the hyperparameters of the baseline methods according to the reported numbers in the papers or in the official implementations.
> - According to the reviewer's comment, we will emphasize in the revised manuscript that each client should have enough computational resources to train a diffusion model.
>   - We originally thought this was reasonable without note, since we were targeting a cross-silo FL setting, where each client is typically organization or datacenter, with sufficient resources and data samples. (cf., Table 1 of [4])
> - We will amend the statement on the compliance with HIPAA.
>   - The original claim of HIPAA compliance in the broader impact statement was to highlight `CooDI`’s potential to support privacy-preserving data sharing in domains governed by privacy regulations (e.g., GDPR and HIPAA), rather than asserting formal legal compliance. Especially in cross-silo FL setting, the data sharing is prohibited due to constraints by confidentiality or legal constraints [4], as `CooDi` targeted for cross-silo FL setting, we considered this perspective.
>
> ----------
> We would like to sincerely, deeply thank **reviewer oT3q** for all the valuable comments in this review.
> Please kindly let us know if any further clarification is needed.
>
>
> [1] Reduce, Reuse, Recycle: Compositional Generation with Energy-Based Diffusion Models and MCMC (Du et al., 2023)
> [2] Compositional Visual Generation with Composable Diffusion Models (Liu et al., 2022)
> [3] How Diffusion Models Learn to Factorize and Compose (Liang et al., 2024)
> [4] Advances and Open Problems in Federated Learning (Kairouz et al., 2019)
> [5] Synthetic data shuffling accelerates the convergence of federated learning under data heterogeneity (Li et al., 2023)
> [6] A Field Guide to Federated Optimization (Wang et al., 2021)
> [7] On the Convergence of FedAvg on Non-IID Data (Li et al., 2019)
> [8] Federated Optimization in Heterogeneous Networks (Li et al., 2018)
> [9] Dimension-free log-sobolev inequalities for mixture distributions (Chen, Chewi and Niles-Weed, 2021)

---

> > ### Comment · Reviewer_oT3q · 2025-08-06
> >
> > Thanks for your response, the clarifications of the term "well trained" was helpful. I realize that it is late into the response session, but I do want to clarify the reason I asked about the naive mixture model. My intention was not to suggest that the naive mixture would be a better model than COODI, I was hoping that a comparison between COODI and the naive mixture could help me better understand the distribution that COODI is sampling from.
> >
> > As of now, I do not feel that the paper does a good job of explaining what distribution the COODI ULA procedure samples from and why that is a desirable distribution. Section 3.3 seems to describe that COODI samples from a diffusion process where the target is a mixture of the targets learned by each local model and the mixture weights depend on the likelihood of the sample as it evolves under each model. This is unclear to me, and it would be helpful to have more clarification. Are these weights optimal in some sense, how were they derived? A more intuitive explanation of this distribution would help motivate COODI and explain why it was developed.

---

> ### Comment · Area_Chair_vjuv · 2025-08-06
>
> Dear Reviewer oT3q，
>
> Thanks for your review. Please read the rebuttal by the authors and see whether your concerns have been sufficiently addressed. If you still have questions you can raise asap such that the authors still have time to response.
>
> AC

---

> ### Author Response · Authors · 2025-08-07
>
> We thank the reviewer for the thoughtful follow-up, and it's not even late since we happily have extended discussion deadline!
> Below, we clarify the **justification on the target distribution** and **clarification on mixture weights**, along with their motivation and derivation.
>
> ### (Recap) what distribution the `CooDI` samples from, resorting to ULA?
> - `CooDI` ultimately aims to sample from **a mixture of heterogeneous and inaccessible local distributions** (lines 37-38; lines 133-134).
> $$p^\star (\boldsymbol{x}) = \sum\nolimits_{i=1}^K w_i p_i (\boldsymbol{x}), \quad \text{ for } i\in[K] \text{ clients} $$
>
> ### The mixture assumption is reasonable and canonical in FL
> - In FL, each client observes only a partial and a slice of the full data distribution (**non-IID** scenario [1,2]).
>   - These slices are not identical, but they often overlap. Thus, the global distribution is not any single probability density, but some combination of them.
> - Since we cannot access raw data or global distribution in FL, the weighted mixture of local distributions (accessible from each client) serves as a practical and flexible proxy for the global target.
>   - This assumption also **aligns with the canonical FL objective** in parameter space—i.e., minimizing a weighted average of local losses [1,2].
> - Accordingly, we model it as a **weighted combination of local distributions**. Note that equal weighting implies all clients are equally informative, which is rarely the case in FL.
>
> ### Taking gradient is what we need to derive the weights in `CooDI`
> - To generate sample from the probability density using ULA, we need **a score of the density**, i.e., a gradient of log probability.
> - As we want to sample from the global distribution, we get **a global score by simply taking gradient** w.r.t. $\boldsymbol{x}$ as follows (i.e., Eq. (8)).
> $$\nabla_\boldsymbol{x} \log p^\star (\boldsymbol{x}) = \sum\nolimits_{i=1}^K \tilde{w_i} \nabla_\boldsymbol{x} \log p_\boldsymbol{\theta_i}(\boldsymbol{x}), \quad \tilde{w_i} = \frac{w_i \exp(-\lambda f_{\boldsymbol{\theta_i}}(\boldsymbol{x}))}{\sum_{j=1}^K w_j \exp(-\lambda f_{\boldsymbol{\theta_j}}(\boldsymbol{x}))}$$
> - To obtain **the weights, $\tilde{w_i}$**,
> - 1. As we stated above, we set $w_i \propto n_i$ .
> - 2. We need to estimate $\exp(-\lambda f_{\boldsymbol{\theta_i}}(\boldsymbol{x}))$.
>   - **Lemma 4.5**) we **theoretically provided that $\lambda=2$ is optimal** for diffusion-compatible ULA sampling.
>   - **Definition 3.1**) we introduced **L2 norm-driven energy-based parameterization trick** to calculate $f_{\boldsymbol{\theta_i}}(\boldsymbol{x})\equiv f_{\boldsymbol{\theta_i}}(\boldsymbol{x_t},t)=\frac{1}{2} \Vert \boldsymbol\epsilon (\boldsymbol{x_t},t) \Vert_2^2$, simply a squared L2-norm of diffusion model's output.
>   - Note that $\exp(-\lambda f_{\boldsymbol{\theta_i}}(\boldsymbol{x})) \propto p_{\boldsymbol{\theta_i}}(\boldsymbol{x})$ (line 138), a likelihood, represents **how plausible the current synthetic sample is**.
> - To sum up, we derive the mixture weights in straightforward manner, jusy by _explicitly taking gradients_, along with _our theoretical result_ and _reparametrization trick_ introduced.
>
> ### Intuitive explanation on the derived weights and distribution
> - The distribution to sample from is **same as the very first mixture distribution**.
> - As the reviewer nicely put:
>   - > ... the mixture weights depend on the likelihood of the sample as it evolves under each model.
> - Indeed, the dynamic weights $\tilde{w}_i$ reflect **how plausible the current synthetic sample is** for each local diffusion model. This guarantees the diversity and quality of  samples from CooDI.
> - This lets ULA explore multiple modes of the global mixture (i.e., inaccessible and heterogeneous local distributions) more effectively than fixed weights, e.g., down-weighting samples from low-likelihood region.
>   - Note that the equally weighted mixture is naturally included as a special case if the energy terms are constant across clients.
> - Again, `CooDI` **adapts mixture weights dynamically**, enabling the collaborative ULA to **explore between modes of local distributions, yielding diverse samples**.
>   - Interestingly, this derived result coincidentally matches _the Exponentiated Gradient update_ [3], which is also a classical adaptive algorithm.
>
> ---
> We hope this clarifies why the mixture assumption is appropriate, how our mixture weights are derived, and how they contribute to the expressiveness and effectiveness of `CooDI` in realistic federated settings.
> We will spare more space to elaborate this perspective in the revised manuscript.
>
> Thank you again for your valuable feedback.
>
>
>
> ### References
> [1] Communication-efficient learning of deep networks from decentralized data (McMahan et al., 2017)
> [2] A Field Guide to Federated Optimization (Wang et al., 2021)
> [3] Exponentiated Gradient versus Gradient Descent for Linear Predictors (Kivinen and Warmuth, 1997)

---

> > ### Comment · Reviewer_oT3q · 2025-08-08
> >
> > Thank you for this response, it was very helpful context for the work. I have increased my score on the assumption that the final text will be updated with more of this context so that it is more clear for readers.

---

> ### Author Response · Authors · 2025-08-08
>
> Dear **reviewer oT3q**,
> We are sincerely delighted to learn that our response was favorable.
> We will **dedicate a separate section to this context in the revised manuscript** to help audiences better understand the context you mentioned.
> We would like to express our gratitude once more for your time and for your understanding of our rebuttal.
>
> Sincerely,
> Authors of Paper 25315

---

### Official Review · Reviewer_d6e9 · 2025-07-05

**Clarity:** 3
**Significance:** 2
**Originality:** 2
**Rating:** 4
**Confidence:** 3

**Summary:**

The core idea of ​​this paper is to regard the problem of federated synthetic data generation as a task of "collaborative sampling from a target distribution mixed from all client local distributions". Instead of exchanging or aggregating model parameters, the server and each client repeatedly exchange noisy prediction scores (energy-based scores) on a small number of samples, and use the unadjusted Langevin algorithm (ULA) to iteratively sample on the global mixed distribution, thereby efficiently and theoretically provably generating high-quality synthetic data in a non-IID environment, while greatly reducing communication overhead and naturally supporting differential privacy.

**Questions:**

1, If the data differences between clients are too large, will COODI still work?

2, COODI uses the mean aggregation result. If it is attacked by malicious attacks, how should COODI respond?

3, At the beginning of the iteration, when the samples are still almost pure noise, the gradient given by the diffusion model is very weak, resulting in a small update amount for each step of ULA, low information content, and the lowest sampling efficiency. How does COODI overcome this problem?

4, If the quality of the diffusion model on each client is poor, is it possible for COODI to improve the quality of the server-side model?

**Ethical Concerns:**

["NO or VERY MINOR ethics concerns only"]

**Limitations:**

1, Each client needs to have a well-trained diffusion model

2, Each client needs to be able to respond in time, which is difficult to promote to large-scale cross-device

3, Although the communication burden is reduced, multiple ULA updates must still be performed on the server side in each round, and noise prediction must be run on the client side. The overall computational cost is still not negligible when the number of nodes is large.

4, The performance differences of each client diffusion model will directly affect the accuracy of the hybrid scoring. If a node model is too underfit or has too large a deviation, it will drag down the global generation quality.

5, The sampling efficiency of ULA is sensitive to the step size. If the step size is too small, the convergence is slow, and if the step size is too large, it may deviate from the target distribution. The tuning cost cannot be ignored.

**Quality:**

3

**Strengths And Weaknesses:**

The quality of this paper is acceptable. The proposed COODI aims to significantly reduce communication overhead and provides theoretical support. I think the clarity of this paper still has room for improvement, especially for some beginners and laymen, it may not be intuitive enough and difficult to understand, but the main method part of the paper is clearly explained. The importance of this paper is moderate, because COODI requires a fully fitted diffusion model to be trained locally, which may bring huge costs under conditions of large data or multiple nodes; when the quality of data from different clients varies greatly, the usability of COODI still needs to be considered; the initial information entropy of the diffusion model is low, and the use of COODI at this time may cause a waste of bandwidth. In addition, the simple weighted average of the energy of each client is also a problem. The originality of this paper is average, and the idea of ​​using a diffusion model to generate synthetic data from GAN is not completely novel.

---

> ### Author Rebuttal · Authors · 2025-07-30
>
> Dear  **reviewer d6e9**,
> We sincerely appreciate the reviewer’s time and effort, and we have carefully addressed each question in detail.
>
> ----------
> ### 1) Case of extreme data heterogeneity
> We appreciate the reviewer’s insightful question regarding `CooDi`’s performance under extreme data heterogeneity.
> - In cases of significant data heterogeneity across clients, the ULA may require additional iterations to effectively mix disparate local density distributions, potentially impacting sample fidelity in extreme scenarios.
> - However, such extreme heterogeneity is rare in practical FL settings, where client data distributions typically exhibit **bounded dissimilarity** rather than arbitrary differences (cf. Assumption 7 in [1], Assumption 3 in [2], Assumption 1 in [3]). This ensures that local distributions, while varied, remain within reasonable bounds.
> - Theoretically, `CooDi` assumes **bounded pairwise chi-squared divergences between local distributions** (Lemma C.6 and Theorem 1 in [4]), aligning with the _bounded dissimilarity_ principle expressed in probabilistic terms.
>   - Theorem 4.6 further indicates that large distributional differences necessitate smaller ULA step sizes, increasing the number of iterations required for convergence.
>   - Empirically, `CooDi` demonstrates robust performance even under heterogeneous data settings, as evidenced by our results (lines 252–253, Tables 1 and 2). These findings confirm that CooDi maintains acceptable sample quality and convergence in practical FL scenarios.
>
> ### 2) Possible attack and defense scenario
> We appreciate the reviewer’s concern regarding the potential vulnerability of `CooDi`’s mean aggregation to malicious attacks.
> - As we specified the target setting as **cross-silo FL setting** (line 56; lines 278-281), `CooDi` assumes **a moderate number of reliable, and known, and addressable clients** [5].
>   - Consequently, our initial design did not prioritize malicious attack scenarios, as cross-silo settings typically involve trusted participants.
> - Nevertheless, we acknowledge that mean aggregation can be susceptible to adversarial manipulations, such as scaling attacks where a malicious client submits inflated or deflated scores.
>   - `CooDi` can successfully mitigate such risks through its theoretically grounded design. Specifically, Lemma 4.4 ensures non-expansive updates in the $\ell_2$-norm, driven by **theoretically-guaranteed step sizes** (Lemma 4.5, Theorem 4.6).
>   - This property allows the central server to detect anomalous updates by simply comparing the norm of a client’s scores against those of others, effectively limiting the impact of scaling attacks.
> - To further enhance robustness, we plan to explore advanced aggregation techniques, e.g., trimmed mean or median-based methods as part of our future work, to address a broader range of practical adversarial scenarios.
>
> ### 3) Concern on early sampling stage
> We thank the reviewer for highlighting the challenge of sampling efficiency in `CooDi`’s early ULA iterations, where noisy samples can lead to weak gradients and smaller updates.
> - To address this, `CooDi` employs a **theoretically derived optimal step size** (Theorem 4.6), which ensures **non-asymptotic convergence** in practice.
>   - Our empirical results (e.g., Tables 1 and 2) are also based on this derived step size, demonstrating robust performance despite early-stage noise.
> - To further enhance convergence speed and sampling efficiency, we propose adopting advanced sampling methods, e.g., the Metropolis-adjusted Langevin Algorithm (MALA [6]) or Hamiltonian Monte Carlo (HMC [7]) in Limitations & Discussion section. (lines 282–284)
>   - We believe these methods could easily reduce the impact of weak gradients and accelerate convergence, but require major changes in design of `CooDi`.
> - We plan to investigate these approaches in future work.
>
> ### 4) Effects of local diffusion models on the quality
> We thank the reviewer for emphasizing the importance of local diffusion model quality in `CooDi`’s collaborative framework.
> - As noted in Remark 4.2, Eq. (9) and (10), and Lemma 4.3, the success of `CooDi` relies heavily on **well-trained** local diffusion models to ensure high-fidelity synthetic data generation.
>   - To support this in practice, we propose an add-on protocol, such as zero-knowledge proof (line 280), to verify that clients faithfully train their local models, enhancing the reliability of the collaborative ULA.
> - Additionally, we clarify that `CooDi` **does not produce a server-side model**, as illustrated in Figure 1, which provides an overview of our federated synthetic data generation process.
> - Instead, `CooDi` aggregates local score estimates to generate server-side synthetic data without centralizing model parameters.
>   - We will update the paper to further emphasize the importance of local model quality and the absence of a server-side model for clarity.
> ----------
> We are grateful to **reviewer d6e9** for constructive feedback.
>
> [1] A Field Guide to Federated Optimization (Wang et al., 2021)
> [2] On the Convergence of FedAvg on Non-IID Data (Li et al., 2019)
> [3] Federated Optimization in Heterogeneous Networks (Li et al., 2018)
> [4] Dimension-free log-sobolev inequalities for mixture distributions (Chen, Chewi and Niles-Weed, 2021)
> [5] Advances and Open Problems in Federated Learning (Kairouz et al., 2019)
> [6] Comments on “Representations of knowledge in complex systems” (Besag, 1994)
> [7] Monte Carlo implementation (Neal, 1996)

---

> ### Comment · Area_Chair_vjuv · 2025-08-06
>
> Dear Reviewer d6e9，
>
> Thanks for your review. Please read the rebuttal by the authors and see whether your concerns have been sufficiently addressed. If you still have questions you can raise asap such that the authors still have time to response.
>
> AC

---

### Decision · Program_Chairs · 2025-09-17

**Decision:**

Accept (poster)

**Comment:**

All reviewers acknowledged that this paper innovatively formulate the task of federated synthetic data generation as sampling from a mixture of decentralized distributions at client sides, and I also found this problem-reframing innovation really exciting. In this way, the underlying problem setting becomes identical to “compositional generation with energy-based diffusion models” [Du et al, 2023] (i.e. [13] in the submission) and the sampling procedure can be directly implemented using the unadjusted Langevin algorithm proposed in [Du et al, 2023].

All reviewers are positive to recommend this paper. But from the wording of Reviewer 1TqA, there might exist some misunderstandings, for example:

> The authors establish a clear connection between diffusion models and energy-based models (EBMs), leveraging this relationship to implement Unadjusted Langevin Algorithm (ULA) sampling without requiring access to model parameters or normalized probability densities. This interpretation allows the system to operate within a well-understood probabilistic framework.

In my understanding, the abovementioned contributions seem rephrased from [Du et al, 2023]. If so, the authors are requested to clearly state that the connection between diffusion models and energy-based models is rephrased from the reference and clearly cite the literature in the Preliminary subsections to avoid misunderstanding.

Another common concern is the practicability of this approach. Although the reformulation of the problem from a new perspective provides a new approach for federated synthetic data generation, this approach requires each client to train a“well trained” local diffusion model and involve frequent client-server communications. It would be more convincing if the authors can find a practical scenario that fits this problem setting.